# Hyperactivation of Nrf2 in early tubular development induces nephrogenic diabetes insipidus

Takafumi Suzuki[1], Shiori Seki[1], Keiichiro Hiramoto[1], Eriko Naganuma[1], Eri H. Kobayashi[1], Ayaka Yamaoka[1], Liam Baird[1], Nobuyuki Takahashi[2], Hiroshi Sato[2] & Masayuki Yamamoto[1,3]

NF-E2-related factor-2 (Nrf2) regulates cellular responses to oxidative and electrophilic stress. Loss of Keap1 increases Nrf2 protein levels, and Keap1-null mice die of oesophageal hyperkeratosis because of Nrf2 hyperactivation. Here we show that deletion of oesophageal Nrf2 in Keap1-null mice allows survival until adulthood, but the animals develop polyuria with low osmolality and bilateral hydronephrosis. This phenotype is caused by defects in water reabsorption that are the result of reduced aquaporin 2 levels in the kidney. Renal tubular deletion of Keap1 promotes nephrogenic diabetes insipidus features, confirming that Nrf2 activation in developing tubular cells causes a water reabsorption defect. These findings suggest that Nrf2 activity should be tightly controlled during development in order to maintain renal homeostasis. In addition, tissue-specific ablation of Nrf2 in Keap1-null mice might create useful animal models to uncover novel physiological functions of Nrf2.

[1] Department of Medical Biochemistry, Tohoku University Graduate School of Medicine, 2-1 Seiryo-machi, Aoba-ku, Sendai 980-8575, Japan. [2] Department of Clinical Pharmacology and Therapeutics, Tohoku University Graduate School of Pharmaceutical Sciences Sendai, 980-8578, Japan. [3] Tohoku Medical-Megabank Organization, 2-1 Seiryo-machi, Aoba-ku, Sendai 980-8575, Japan. Correspondence and requests for materials should be addressed to T.S. (email: taka23@med.tohoku.ac.jp) or to M.Y. (email: masiyamamoto@med.tohoku.ac.jp).

Accumulating lines of evidence support the notion that oxidative and electrophilic stresses form the molecular basis for many diseases, including cancer, diabetes and inflammation[1]. Our bodies are equipped with a number of defense systems that can protect us from these damaging insults. In addition to the natural induction of cytoprotective systems, activation of these defense systems using small-molecule chemical inducers is considered to be an effective strategy for the treatment and/or prevention of disease. To this end, appropriate model animal systems that allow us to evaluate the physiological consequences of an increase in the expression of cytoprotective systems are urgently required.

Transcription factor Nrf2 (NF-E2-related factor-2) is essential for the oxidative and electrophilic stress responses of animals[1]. Downstream target genes of Nrf2 include enzymes that act in the antioxidant and detoxification pathways, which regulate the cellular adaptation to oxidative and xenobiotic stresses[2]. The *Nrf2* gene knockout mouse clearly demonstrates that Nrf2 plays an important role in the response to oxidative and electrophilic stresses[3,4]. Under homeostatic and stress-free conditions, cellular Nrf2 abundance is maintained at a very low level as the ubiquitin E3 ligase complex composed of Keap1 and Cullin 3 (Cul3) specifically promotes the ubiquitination and proteasomal degradation of Nrf2 (refs 5,6). Notably, Keap1 acts as a sensor for electrophilic and oxidative stresses by using reactive cysteine residues within the protein[2,7]. Exposure to electrophiles or reactive oxygen species (ROS) hampers Keap1 activity, reducing Nrf2 ubiquitination and leading to the stabilization and nuclear translocation/accumulation of Nrf2 (ref. 2). Subsequently, the expression of a battery of Nrf2 target genes is induced for cytoprotection against these insults. Thus cellular Nrf2 activation results from the repression of Keap1-mediated Nrf2 degradation[1,2].

In order to investigate the function of Nrf2 in mammals, we generated Keap1-null mice in which Nrf2 is constitutively activated, and Nrf2 target genes are upregulated throughout the whole body[8]. However, *Keap1*-null mice are juvenile lethal due to hyperkeratosis in the upper digestive tract, which leads to the obstruction of the oesophagus and death by starvation. Importantly, the lethality of the *Keap1*-null mice is cancelled by simultaneous deletion of the *Nrf2* gene[8], indicating that the *Keap1*-null phenotype is a consequence of the constitutive activation of Nrf2. Knockdown of the *Keap1* gene due to floxation of the Keap1 locus with loxP sites leads to the constitutive accumulation of Nrf2 throughout the body without lethality[9]. However, in this case, Nrf2 activation is weaker than that of the Keap1-null mice due to the partial retention of Keap1 expression[9]. Tissue-specific disruption of the *Keap1* gene using the Cre-loxP system has been used to investigate the physiological effects of full activation of Nrf2 by the complete deletion of Keap1 (refs 9–12). However, there are limitations in the availability of mouse lines expressing the Cre recombinase in a tissue-specific manner.

To generate a viable mouse model harbouring systemic full activation of Nrf2, we decided to rescue the lethality of the global Keap1-knockout mouse using *Keratin5*-Cre mice (K5-Cre), which express Cre recombinase in squamous epithelium under regulation of the *Keratin5* promoter[13]. The squamous epithelium-specific disruption of the *Keap1* gene using K5-Cre mice recapitulates the lethality of *Keap1*-null mice[9], indicating that Nrf2 accumulation in the squamous epithelium is responsible for the lethality of systemic *Keap1*-null mice. In contrast, a squamous epithelium-specific Nrf2 deficiency in the context of systemic Keap1-deficient mice (*Keap1*[−/−]::*Nrf2*[Flox/Flox]::K5-Cre, referred to as Nrf2-deficient in oesophagus and *Keap1*-null mice, NEKO) corrects the hyperkeratosis of the oesophagus and subsequent lethality, while full activation of Nrf2 is observed in most tissues, with the exception of the oesophagus and skin. Through the analysis of NEKO mice, we found a novel phenotype in the kidney, which is attributed to the full activation of Nrf2 by complete deletion of Keap1.

## Results

**Oesophageal Nrf2 deletion rescues Keap1-null mouse lethality.** To remove Nrf2 expression in the oesophagus of systemic *Keap1*-null mice, we utilized *K5-Cre*[13], which expresses the Cre recombinase in squamous epithelium, and generated *Keap1*[−/−]::*Nrf2*[Flox/Flox]::*K5-Cre* (NEKO) mice. As expected, *Nqo1* and *Gclc*, prototypical Nrf2 target genes[2], were upregulated in most tissues of *Keap1*[−/−]::*Nrf2*[Flox/Flox] mice, whereas *Nqo1* and *Gclc* induction was reduced in the oesophagus and skin but not in the other tissues in NEKO mice (Fig. 1a and Supplementary Fig. 1). Consistent with the *Nqo1* and *Gclc* expression, the hyperkeratosis of the oesophagus and forestomach observed in *Keap1*[−/−]::*Nrf2*[Flox/Flox] mice was rescued in NEKO mice (Fig. 1b–e and Supplementary Fig. 2a–c). These observations indicate that NEKO mice display constitutive activation of Nrf2 in most tissues except the squamous epithelium. In good agreement with the improvement of hyperkeratosis, the juvenile lethality of *Keap1*[−/−]::*Nrf2*[Flox/Flox] mice was rescued. However, NEKO mice displayed significantly poor survival compared with control mice (Fig. 1f). In addition, NEKO mice showed growth retardation (Fig. 1g), although blood glucose level was normal (Supplementary Fig. 3). As these phenotypes have not been observed in mice with systemic double deficiency of *Keap1* and *Nrf2* genes[8], we surmise that the poorer survival and growth retardation are caused by Nrf2 activation in tissues other than the squamous epithelium.

**NEKO mice display polyuria with low osmolality.** We found that NEKO mice display a significant increase in urine volume (Fig. 2a,b). The polyuria was not observed in *Keap1*[−/−]::*Nrf2*[−/−] mice (Fig. 2c), indicating that this is an Nrf2-dependent phenotype. In addition, knockdown of Keap1, which causes mild activation of Nrf2 due to a reduction in the expression of Keap1 throughout the whole body[9], also did not cause polyuria (Fig. 2d). These observations indicate that the polyuria phenotype requires the complete loss-of-Keap1 activity and full activation of Nrf2. Consistent with the light coloured urine of NEKO mice, the urine osmolality was drastically decreased (Fig. 2e) and intake of drinking water was increased (Fig. 2f), without an effect on food consumption (Fig. 2g). These results indicate that NEKO mice suffer from excessive thirst and the excretion of a large amount of dilute urine, suggesting that NEKO mice cannot make concentrated urine.

**Polyuria in NEKO mice is not due to a neurological defect.** Water reabsorption is regulated by the antidiuretic hormone, vasopressin (AVP), which is produced in the hypothalamus[14]. Therefore, to clarify whether polyuria in NEKO mice is provoked by a neurological defect, we examined AVP production in NEKO mice. However, the plasma AVP level and *Avp* gene expression in the hypothalamus of NEKO mice were relatively elevated rather than decreased when compared with those of control mice (Fig. 3a,b). In addition, neuron-specific *Keap1* knockout mice (*Keap1*[Flox/Flox]::*Nestin*-Cre, Keap1-NKO) did not display polyuria (Fig. 3c). These results indicate that it is unlikely that a neurological defect of AVP production causes the defect in making concentrated urine in NEKO mice.

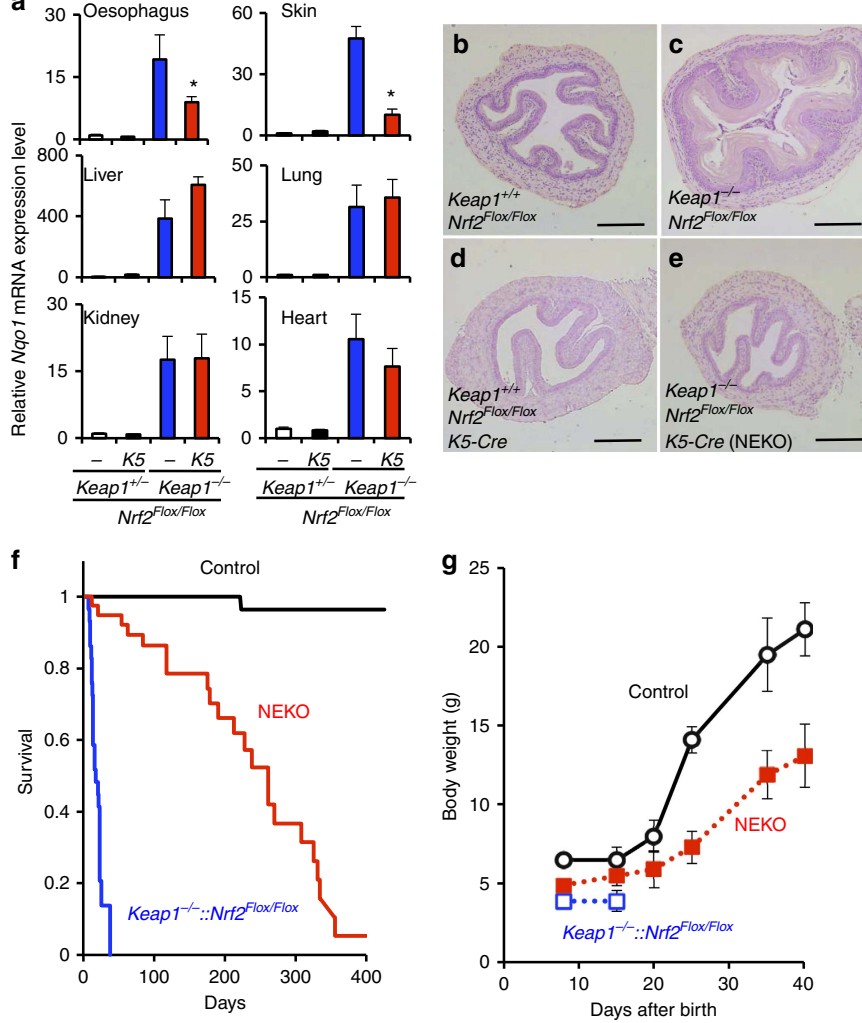

**Figure 1 | Generation of a mouse model that lacks Keap1 throughout the whole body and Nrf2 in the squamous epithelium.** (**a**) Relative *Nqo1* gene expression level compared with *Hprt* expression in the oesophagus, skin, liver, lung, kidney and heart of mice at 10 days of age. Data are the means ± s.e. ($n=4$). (*$P<0.05$ compared with $Keap1^{-/-}::Nrf2^{Flox/Flox}$, unpaired *t*-test) (**b–e**) Representative images of haematoxylin and eosin staining of transverse sections of the oesophagus from mice at 10 days of age ($n=3$). Scale bar, 100 μm. (**f**) Survival curve for control ($n=56$), $Keap1^{-/-}::Nrf2^{Flox/Flox}$ ($n=29$) and NEKO ($n=42$) mice. (**g**) Growth curve for control ($n=15$), NEKO ($n=10$) and $Keap1^{-/-}::Nrf2^{Flox/Flox}$ ($n=16$) mice. Data are the means ± s.e.

**Reduction of AQP2 protein level in the kidney of NEKO mice.**
It is well established that aquaporin 2 (AQP2) is a water channel responsible for the reabsorption of water in the kidney[15,16]. Therefore, we next examined the AQP2 protein level in the kidney of NEKO mice (Fig. 3d,e). To our surprise, we found that AQP2 protein, especially its glycosylated form (Fig. 3d, open arrowhead), was significantly decreased in the kidney of NEKO mice (Fig. 3e), suggesting that apical membrane trafficking and shedding of AQP2 could be stimulated. In contrast, the *Aqp2* mRNA level was unchanged in the kidney of NEKO mice (Fig. 3f), indicating that the reduction in the AQP2 protein level is caused by a post-transcriptional event. Reduced protein level of AQP2, but not SLC34A1, a marker for proximal convoluted tubules, was observed as early as 10 days of age (Supplementary Fig. 4a), while the *Avp* mRNA level was not changed substantially in the hypothalamus of NEKO mice at 10 days of age (Supplementary Fig. 4b). These observations indicate that AQP2 reduction in the kidney precedes the increase in AVP, suggesting that the increase in AVP is a compensatory response secondary to the reduction of AQP2.

Showing very good agreement with the western blotting analysis, immunostaining of AQP2 in the kidney showed weaker signals in the cytoplasm of collecting ducts in NEKO mice (Fig. 3h) compared with those in the control mice (Fig. 3g). AQP2 accumulation in the apical membrane of the collecting duct was observed in NEKO mice (Fig. 3h, inset), suggesting that compensatory translocation of AQP2 to the luminal membrane and excretion of AQP2 to the urine is enhanced, and/or AQP2 is degraded inside the cells. The weaker signals and apical translocation of AQP2 were observed in NEKO mice even at 10 days of age (Supplementary Fig. 5a,b). Of note, the expression levels of AQP4, another water channel expressed in collecting ducts, and NCC, a distal convoluted tubule marker, were not affected in NEKO mice (Supplementary Fig. 5c–f). These results indicate that the water reabsorption defect in NEKO mice is caused by a reduction in the AQP2 protein level in the kidney, leading to a decrease in the ability of the kidney to concentrate urine by removing water.

**Renal pelvis pressure due to polyuria in NEKO mice.** Missense mutations in the *AQP2* gene are frequently found in congenital nephrogenic diabetes insipidus (NDI) patients[17]. Consistently, mutations that lead to loss-of-AQP2 function in mice recapitulate NDI and result in hydronephrosis[18,19]. Therefore, we looked at

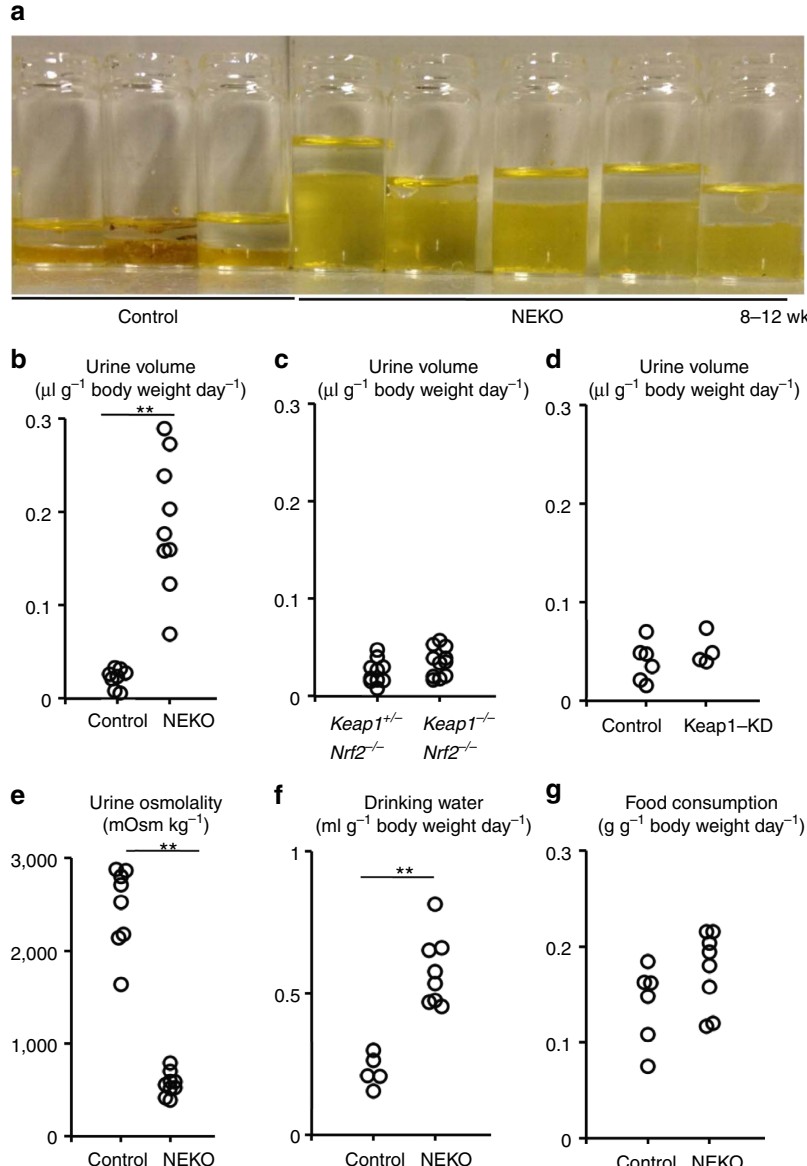

**Figure 2 | NEKO mice display polyuria.** (**a**) The representative urine appearance of control and NEKO mice. (**b–d**) Urine volume of NEKO (**b**), *Keap1*$^{-/-}$*::Nrf2*$^{Flox/Flox}$ (**c**) and Keap1-KD (**d**) mice at 8–12 weeks of age. (**e**) Urine osmolality of control and NEKO mice. (**f**) Volume of drinking water consumed by control and NEKO mice. (**g**) Food consumption of control and NEKO mice. (**P < 0.01, unpaired *t*-test).

the kidney of NEKO mice at 8 weeks of age or later and found bilateral hydronephrosis characterized by an enlarged pelvic space (Fig. 4a,b and Supplementary Fig. 6a). The dilation of the renal pelvis was not observed at 10 days (Fig. 4c,d) or 4 weeks of age (Supplementary Fig. 6b). As increases in urine volume are already observed at 4 weeks of age (Fig. 4e), these observations suggest that polyuria occurs before the structural change in the kidney.

Indeed, a higher magnification view of kidney histology of NEKO mice at 8 weeks of age showed dilation of the collecting ducts (Fig. 4g, open arrowheads) compared with those of littermate control mice (Fig. 4f), indicating elevated pressure in the kidney. Histology also revealed that, while most glomeruli appeared normal, mildly damaged glomeruli and infiltration of inflammatory cells were observed in the NEKO mouse kidney (Fig. 4h,i), although the plasma creatinine level was normal (Supplementary Fig. 6c). A slight increase in the plasma level of blood urea nitrogen, an indicator of kidney function, was observed in NEKO mice (Fig. 4j). These results suggest that mild kidney damage is likely a consequence of polyuria.

To test the alternative possibility that obstruction of the ureter causes hydronephrosis in NEKO mice, we infused a dye into the kidney pyelocaliceal space. The dye was successfully introduced from the pelvis to the bladder in the control (Fig. 4k) and NEKO (Fig. 4l) mice, and macroscopic observation suggested that there was no ureteral obstruction or dilatation in NEKO mice (Fig. 4m). These observations indicate that NEKO mice first develop NDI, and the resultant increase in urine volume overwhelms the capacity of the ureter to transfer urine from the kidney to the bladder, thereby causing induction of hydronephrosis in the absence of an anatomical obstruction.

**Impaired response to dehydration and AVP**. To further verify the NDI in NEKO mice, we examined their response to dehydration. Blood levels of sodium, potassium and chloride in NEKO mice were significantly increased during dehydration, while those of control mice were unchanged (Fig. 5a–c). In addition, urinary osmolality of NEKO mice was lower than that

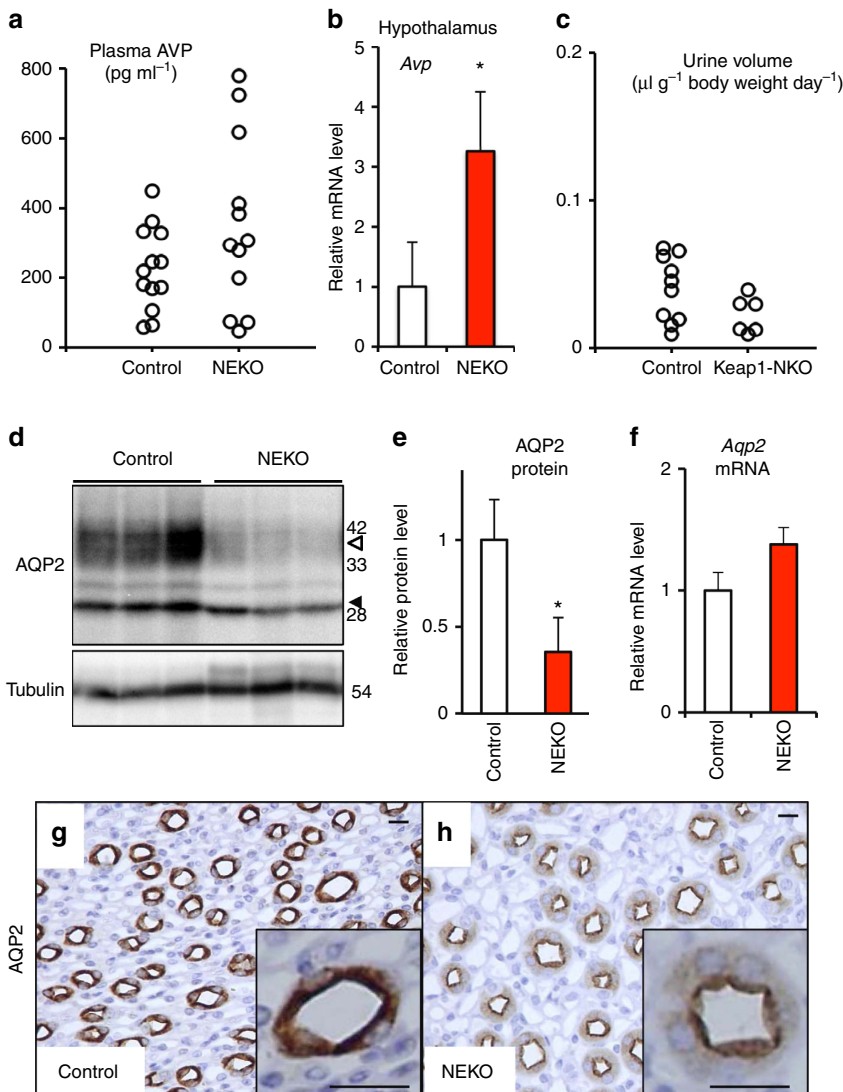

**Figure 3 | NEKO mice display nephrogenic diabetes insipidus.** (**a**) Plasma AVP level in control and NEKO mice. (**b**) Relative *Avp* gene expression level in the hypothalamus of control and NEKO mice. Data are the means ± s.e. ($n = 4$). (*$P < 0.05$, unpaired *t*-test) (**c**) Urine volume of control and Keap1-NKO mice. (**d**) Western blotting analysis of AQP2 protein in the kidney of control and NEKO mice. Open and closed arrowheads indicate glycosylated and non-glycosylated forms of AQP2, respectively. The molecular weight standards are shown on the right. (**e**) Graphical representation of total AQP2 protein level shown in **d**. Data are the means ± s.e. ($n = 3$). (*$P < 0.05$, unpaired *t*-test) (**f**) Relative *Aqp2* gene expression level in the kidney of control and NEKO mice. Data are the means ± s.e. ($n = 7$). (**g,h**) AQP2 immunostaining of kidney sections of control and NEKO mice. Insets show a higher magnification view of the collecting duct. Scale bar, 10 μm.

of control mice even under dehydration (Fig. 5d). These results thus indicate that NEKO mice cannot make concentrated urine in response to dehydration.

To further verify whether the response to AVP is impaired in NEKO mice, we administered mice with desmopressin (dDAVP), a synthetic AVP analogue. We found that dDAVP rapidly increased urine osmolality in control mice but not in NEKO mice (Fig. 5e). This result demonstrates that the response to AVP is impaired in NEKO mice.

There remains the alternative possibility that the AVP–AQP2 system in collecting ducts cannot work properly due to a defect in diluting luminal fluid in the ascending limb of the loop of Henle as in the case of Bartter's syndrome. To address this question, we measured urine osmolality of NEKO mice after water loading. Normally, upon challenge with excess water, AVP secretion is stopped and the kidney makes dilute urine to excrete the water. Indeed, we found that urine osmolality of NEKO mice after water loading was lower than that of the plasma ($< 300 \, \text{mOsm kg}^{-1}$

$H_2O$, Supplementary Fig. 7), indicating that the diluting function is not compromised in NEKO mice. Accordingly, we conclude that polyuria in NEKO mice is due to reduced expression of AQP2 in the collecting duct.

**Nephrogenic diabetes insipidus in NEKO mice.** To examine in which cells Keap1 deletion activates Nrf2, we carried out immunostaining for NQO1, which is a prototypical Nrf2 target gene[2]. We found marked induction of NQO1 expression in the kidney (Supplementary Fig. 8a,b). Higher magnification analysis showed a strong signal for NQO1 in the cortical tubules and collecting ducts but not in the glomeruli (Supplementary Fig. 8c–f). These observations support our contention that Keap1 regulates Nrf2 activity mainly in the renal tubular cells.

To delineate the contribution of Nrf2 activation in the renal tubular cells, we utilized the doxycycline (DOX)-inducible Cre transgenic mouse system and deleted the Keap1 gene specifically

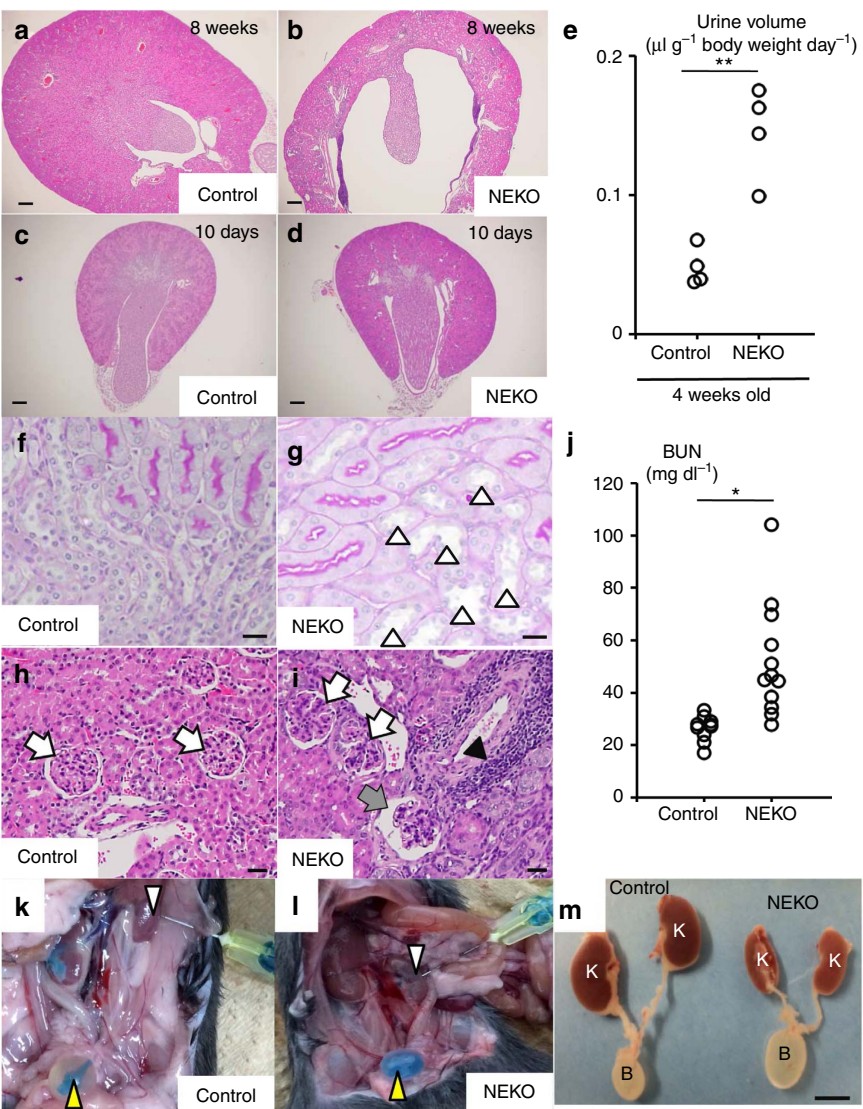

**Figure 4 | NEKO mice display bilateral hydronephrosis.** (**a–d**) Representative images of haematoxylin and eosin (HE) staining of coronal kidney sections of control and NEKO mice at 8 weeks (**a,b**) and 10 days (**c,d**) of age ($n = 4$). Scale bar, 100 μm. (**e**) Urine volume of control and NEKO mice at 4 weeks of age. (**P < 0.01, unpaired t-test). (**f–i**) Periodic acid-Schiff staining of renal outer medulla (**f,g**) and HE staining of renal cortex (**h,i**) of control and NEKO mice at 8 weeks of age. Open and closed arrowheads indicate dilated collecting ducts and inflammatory cell infiltration, respectively. Open and closed arrows indicated normal and damaged glomerulus, respectively. Scale bar, 10 μm. (**j**) Plasma blood urea nitrogen level in control and NEKO mice at 8–12 weeks of age. (**P < 0.01, unpaired t-test). (**k,l**) Ink infused into the pyelocaliceal space of the kidney of control and NEKO mice at 8 weeks of age. (**m**) Urinary bladder (B) and kidney (K) of control and NEKO mice at 8 weeks of age. Scale bar, 5 mm.

in the renal tubular cells. We also generated *Keap1*^*Flox/Flox*^*::Pax8*-rtTA::*tetO*-Cre (referred to as Keap1-TKO) mice and treated the mice with DOX from 4 weeks of age (Fig. 6a,b). Unexpectedly, the Keap1-TKO mice at 8 weeks of age did not show polyuria (Fig. 6a) even though *Keap1* expression was negligible and *Nqo1* expression was highly induced (Fig. 6b). However, when Keap1 deletion was induced from the embryonic stage, by administration of DOX to the pregnant mother, the Keap1-TKO mice at 8 weeks of age displayed a significant increase in urine volume (Fig. 6c), despite the fact that the expression of *Keap1* and *Nqo1* (Fig. 6d) were comparable to those of the Keap1-TKO mice treated with DOX from the adult stage (Fig. 6b). In addition, the Keap1-TKO mice treated with DOX from an embryonic stage had urine with lower osmolality (Fig. 6e) and drank more water than control mice (Fig. 6f), indicating that the Keap1-TKO mice display NDI in a similar way to NEKO mice. These observations suggest the importance of a narrow

developmental window for Nrf2 activation by Keap1 deletion on the subsequent increase in urine volume in Keap1-TKO mice.

Consistent with the increase in urine volume, the kidney of the Keap1-TKO mice showed dilation of the pelvis (Fig. 7a,b) and tubules (Fig. 7c,d). The level of AQP2 protein, especially its glycosylated form, was reduced in the Keap1-TKO mice (Fig. 7e,f), without a corresponding change in the level of the transcript of the *Aqp2* gene (Fig. 7g). The level of AQP2 in the kidney from Keap1-TKO mice treated with DOX during the adult stage was not significantly reduced, although there was a slight decrease in the glycosylated form of AQP2 (Supplementary Fig. 9). Taken together, these results demonstrate that Nrf2 activation by renal Keap1 deletion during development, but not during adulthood, leads to a reduction in the level of the AQP2 protein in the kidney, thereby leading to NDI.

We also found reduced levels of haematocrit and haemoglobin (Supplementary Fig. 10a,b) in NEKO mice, indicating mild

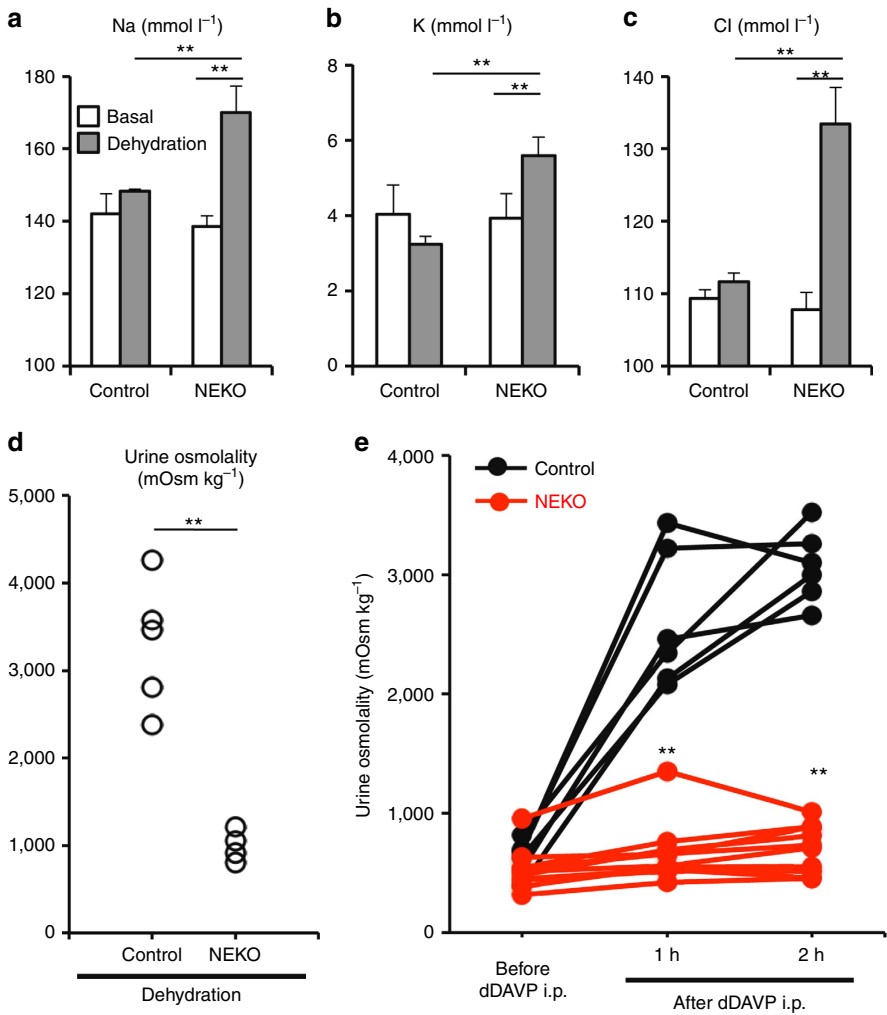

**Figure 5 | NEKO mice have a defect in response to dehydration and vasopressin.** (**a**–**c**) Blood sodium (**a**), potassium (**b**) and chloride (**c**) in control and NEKO mice at 8–12 weeks of age after 24 h of water deprivation. Data are the means ± s.e. ($n = 4$) (**$P < 0.01$, unpaired $t$-test). (**d**) Urine osmolality of NEKO mice at 8–12 weeks of age after 24 h of water deprivation. (**$P < 0.01$, unpaired $t$-test) (**e**) dDAVP stimulation test of control and NEKO mice at 8–12 weeks of age after water loading. Mice were given water containing 5% sucrose before the experiment to ensure production of dilute urine. Urine was collected and 1 ng g$^{-1}$ body weight of dDAVP was i.p. injected. Urine osmolality was measured at 1 and 2 h after the injection. (**$P < 0.01$, unpaired $t$-test).

anaemia in NEKO mice. In addition, the body weight of NEKO mice was lower than that of control mice (Supplementary Fig. 10c). These phenotypes of NEKO mice were not recapitulated in the Keap1-TKO mice. In addition, poor survival of NEKO mice was also not recapitulated in the Keap1-TKO mice (Supplementary Fig. 11). These observations indicate that these phenotypes are independent of renal tubular Nrf2 hyperactivation.

**Enhanced secretion of AQP2 protein to the urine of NEKO mice.** As AVP-mediated trafficking of AQP2 to the apical membrane is known to elicit urinary excretion of AQP2 (ref. 20), we examined urinary AQP2 protein levels. To our surprise, we found that AQP2 protein level in the urine of NEKO mice was significantly increased compared with that of control mice (Fig. 8a,b). As the glycosylated form of AQP2 protein was strikingly reduced in the kidneys of the NEKO and Keap1-TKO mice (Figs 3d and 7e), it seemed plausible that Nrf2 regulates the expression of glycosylation-related genes that are involved in the trafficking of AQP2. To address this issue, we carried out microarray expression analysis of genes in the kidneys of NEKO mice (Supplementary Fig. 12a,b) and searched for glycosylation-related

genes whose expression is changed in NEKO mice. Among the genes increased in NEKO mice, we found upregulation of *Clec4d* and *Clec4n*, members of the C-type lectin family. As some members of the lectin family are involved in subcellular protein trafficking[21], we hypothesized that Clec4d and Clec4n may enhance the transport of AQP2 to the apical membrane and thereby enhance urinary excretion of AQP2. Indeed, transcript levels of *Clec4d* and *Clec4n* genes are already increased in the kidney of NEKO mice at 10 days of age (Fig. 8c), indicating that upregulation of *Clec4d* and *Clec4n* gene expression occurs before the structural change in the kidney.

To examine which cells in the kidney express *Clec4d* and *Clec4n* genes, we separated collecting duct cells from the kidney by biotinylated Dolichos biflorus agglutinin (DBA)/streptavidin magnetic beads[22]. We found that expression levels of *Clec4d* and *Clec4n* genes were upregulated in the collecting duct cells isolated from NEKO mice compared with those isolated from control mice (Supplementary Fig. 13).

As *Clec4d* and *Clec4n* genes are normally expressed in macrophages[23], we examined their expression in bone marrow-derived macrophages (BMDMs). We found that diethylmaleate (DEM), an Nrf2-activating chemical, increased the expression

of these genes in BMDMs in an Nrf2-dependent manner (Supplementary Fig. 14a,b). To address whether *Clec4d* and *Clec4n* are direct target genes of Nrf2, we searched for Nrf2-binding sites around the *Clec4d* and *Clec4n* gene loci using the

chromatin immunoprecipitation sequence (ChIP-seq) data from a previous study[24] and found multiple binding sites for Nrf2 around the *Clec4d* and *Clec4n* gene loci (Supplementary Fig. 14c). We validated Nrf2 binding to these sites by manual ChIP using

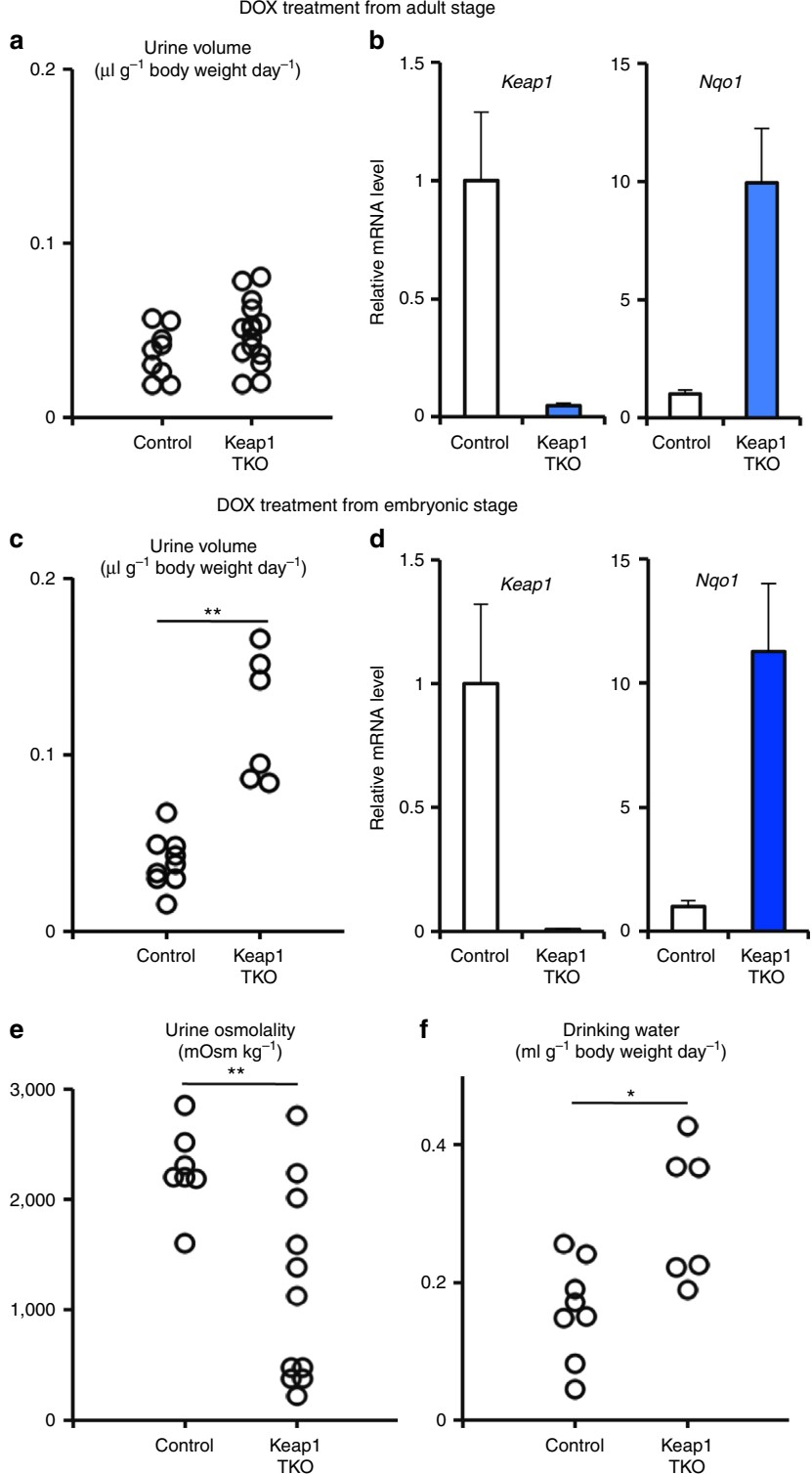

**Figure 6 | Deletion of renal tubular Keap1 from embryonic stage but not from adult stage leads to polyuria.** (**a,b**) Control and Keap1-TKO mice were treated with DOX from adult stage. Urine volume (**a**) and relative transcript levels of *Keap1* and *Nqo1* genes in the kidney (**b**) of the mice are shown. (**c**–**f**) Control and Keap1-TKO mice were treated with DOX from embryonic stage. Urine volume (**c**), relative transcript levels of *Keap1* and *Nqo1* genes in the kidney (**d**), urine osmolality (**e**) and volume of drinking water consumed (**f**) by the mice are shown. Data of gene expression are the means ± s.e. ($n = 8$). (*$P < 0.05$. **$P < 0.01$, unpaired *t*-test).

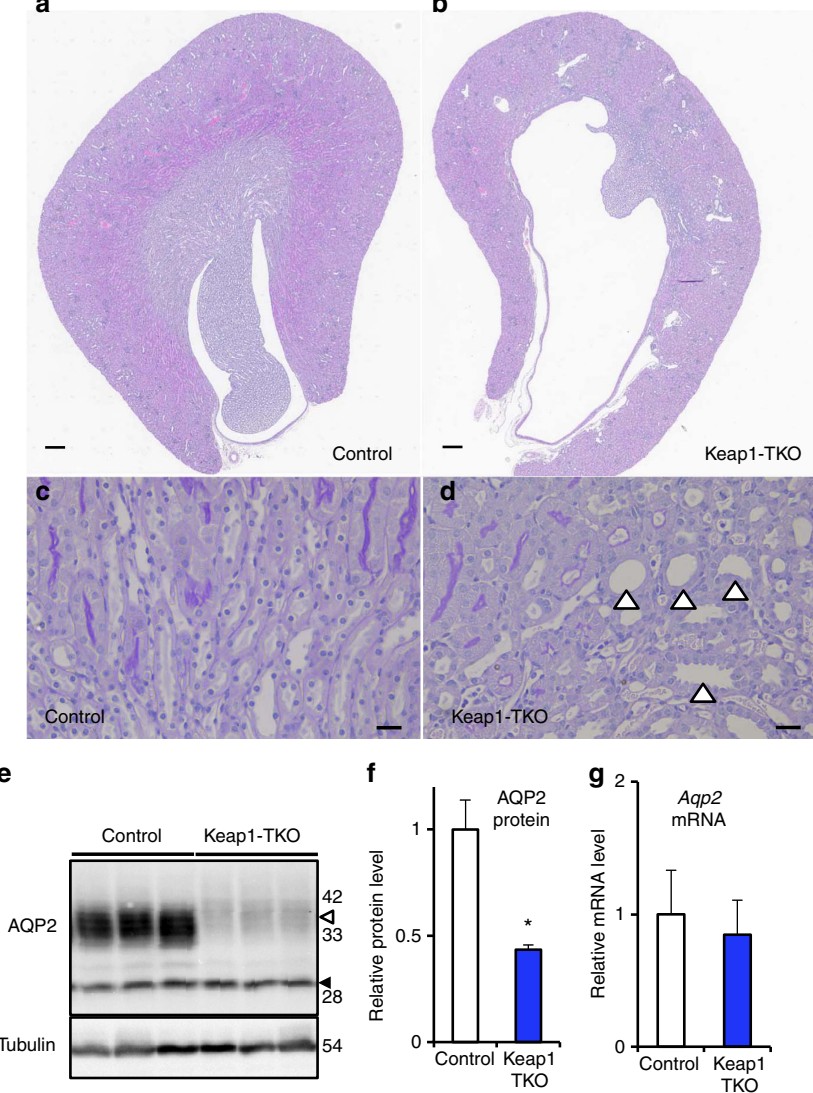

**Figure 7 | Renal tubular Keap1-deficient mice from adult stage recapitulate hydronephrosis and reduction of AQP2 in the kidney.** Representative images of haematoxylin and eosin staining (**a,b**) and periodic acid-Schiff staining (**c,d**) of kidney coronal sections of control and Keap1-TKO mice ($n = 3$). Open arrowheads indicate dilated renal tubules. Scale bars, 100 μm (**a,b**) and 10 μm (**c,d**). (**e**) Western blotting analysis of AQP2 protein in the whole kidney of control and Keap1-TKO mice. Open and closed arrowheads indicate glycosylated and non-glycosylated form of AQP2, respectively. The molecular weight standards are shown on the right. (**f**) Graphical representation of total AQP2 protein level shown in **e**. Data are the means ± s.e. ($n = 3$) (*$P < 0.05$, unpaired *t*-test) (**g**) Relative *Aqp2* gene expression level in the kidney of control and Keap1-TKO mice. Data are the means ± s.e. ($n = 8$).

DEM-treated BMDMs (Supplementary Fig. 14d). These results suggest that *Clec4d* and *Clec4n* genes are direct target genes of Nrf2.

In spite of the hyperactivation of Nrf2 in renal tubular cells, the transcript levels of *Clec4d* and *Clec4n* genes did not increase in Keap1-TKO mice treated with DOX from the adult stage (Fig. 8d). In contrast, an increase in the transcript levels of *Clec4d* and *Clec4n* genes was recapitulated in Keap1-TKO mice treated with DOX from an embryonic stage (Fig. 8e). Of note, the increase in expression levels of the *Clec4d* and *Clec4n* genes shows very good correlation to the urinary concentrating defects. These findings support our contention that Nrf2 hyperactivation in renal tubular cells during development leads to the upregulation of *Clec4d* and *Clec4n* gene expression and enhanced urinary excretion of the AQP2 protein, which consequently causes a reduction in the level of the AQP2 protein in the kidney, ultimately leading to NDI.

## Discussion

In this study, we generated a new mouse model, NEKO, which has high Nrf2 activity due to Keap1 deletion but without juvenile lethality and hyperkeratosis of the upper digestive tract. Surprisingly, through the study of NEKO mice, we found a novel phenotype, namely, polyuria with low osmolality and consequent renal structural damage, presumably due to a reduction of the AQP2 protein in the kidney. The renal phenotypes were recapitulated by renal tubular-specific Keap1 deletion during development but not adulthood, indicating that renal activation of Nrf2 at an early stage is responsible for the polyuria and kidney damage observed in NEKO mice. The reduction of AQP2 protein in the kidney is likely due to dysregulation of AQP2 trafficking provoked by upregulation of *Clec4d* and *Clec4n* genes, members of C-type lectin family, in a developmental stage-specific manner. These results demonstrate for the first time that Nrf2 activation during renal development leads to NDI.

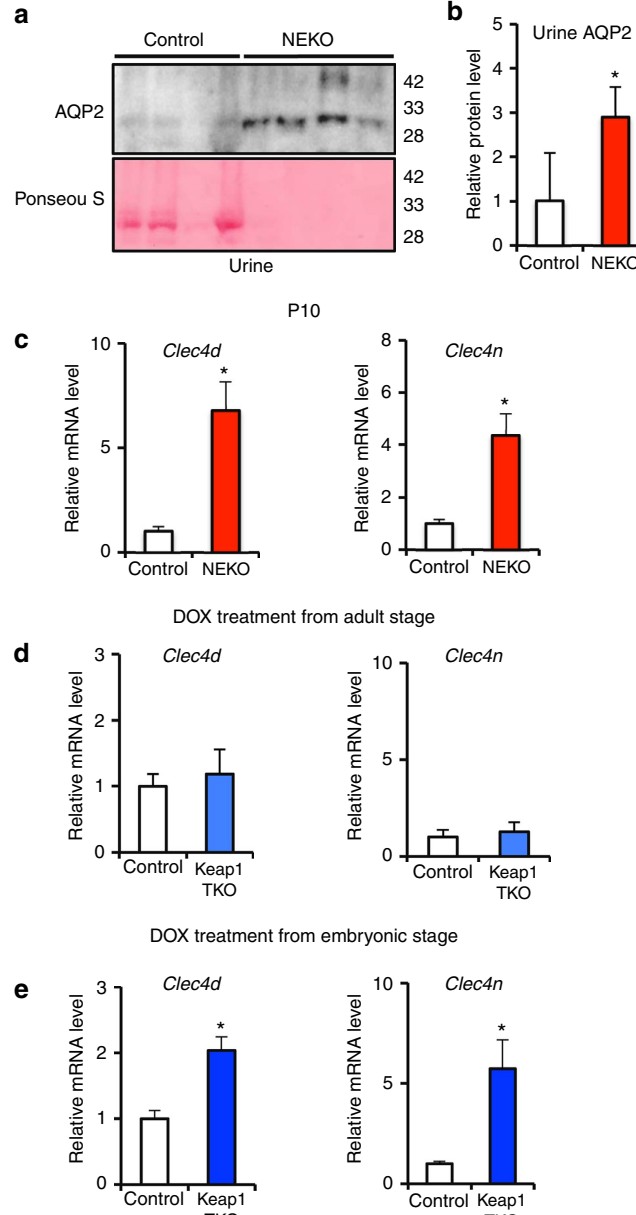

**Figure 8 | Urinary excretion is enhanced in NEKO mice.** (**a**) Western blotting analysis of AQP2 protein in urine of control and NEKO mice. Each sample was loaded with the same amount of creatinine. Ponseau S staining of the membrane is shown. (**b**) Graphical representation of total AQP2 protein level shown in panel **a**. Data are the means ± s.e. (*n* = 4) (*P < 0.05, unpaired *t*-test). (**c–e**) Relative transcript levels of *Clec4d* and *Clec4n*, members of the C-type lectin family genes, in the kidney of control and NEKO mice at 10 days of age (**c**) and control and Keap1-TKO mice treated with DOX from adult stage (**d**) and from embryonic stage (**e**). Data are the means ± s.e. (*n* = 4 for **c**, *n* = 8 for **d**,**e**). (*P < 0.05, unpaired *t*-test).

Keap1 tightly represses Nrf2 activity in normal conditions, and Nrf2 activity is induced by loss of Keap1. Therefore, a *Keap1*-null mouse is a desired model in which to investigate the function of Nrf2. However, due to the juvenile lethality of systemic *Keap1*-null mice[8], there has been a limitation in the phenotypic analysis of adult mice[9]. Generation of NEKO mice in this study makes it possible to analyse the physiological contribution of constitutive Nrf2 activation in adult animals as the mice can survive to adulthood. In this regard, it is interesting to note that

we have previously generated a Keap1 knockdown line of mice, which causes constitutive Nrf2 activation but not lethality[9]. The polyuria was not observed in this Keap1 knockdown line of mice, perhaps as a mild increase in Nrf2 may not be enough to provoke NDI. Thus it is not until NEKO mice were generated that polyuria and hydronephrosis were identified as a consequence of constitutive Nrf2 activation by the complete knockout of Keap1. The murine phenotypes of loss-of-Keap1-mediated Nrf2 hyperactivation, such as oesophageal hyperkeratosis and a urinary concentrating defect, suggests that Nrf2 regulates not only cytoprotective genes such as antioxidant and detoxifying genes, but also other genes involved in cell fate and function for organismal homeostasis. Importantly, although elevations of NRF2 levels have been identified in various types of human cancers[22,25], spontaneous tumorigenesis has not been observed in NEKO mice. These findings indicate that mere hyperactivation of Nrf2 is not sufficient to cause cancer, but may assist cancer growth and drug resistance.

The polyuria in NEKO mice is likely caused by a reduction of AQP2 in the kidney due to the enhanced urinary excretion of AQP2. Interestingly, urinary exosomal AQP2 has been suggested to be a useful marker for the diagnosis of renal disease[26,27]. As Nrf2 target genes include many antioxidant genes[2], one explanation might be that the elimination of endogenous ROS by excessive activation of Nrf2 contributes to AQP2 trafficking. However, as there is no report regarding the relationship between ROS and AQP2, coupled with the fact that our study indicates that a development stage-specific Nrf2 target gene is responsible for the renal phenotype, this suggests that the general induction of antioxidant genes by Nrf2 is unlikely to be responsible for the polyuria phenotype.

It is noteworthy that the glycosylated form of AQP2 is specifically reduced in the kidney, implying that glycosylation of AQP2 is involved in the mechanism of NDI development. Glycosylation of AQP2 is not essential for its tetramerization[28] but is important for AQP2 cell surface localization[29]. Although the involvement of lectins in the process has been suggested[30], it remains unknown which lectin is important for AQP2 trafficking. We found that upregulation of *Clec4d* and *Clec4n* in the kidney is correlated with defects in urinary concentration, suggesting that *Clec4d* and *Clec4n* are strong candidate genes responsible for transporting AQP2 to the apical membrane of renal collecting duct cells. Lectins such as galectins[21] and mannose-6 phosphate receptor[30] are involved in subcellular protein trafficking, although to date there has been no report regarding the trafficking function of Clec4d and Clec4n. Our findings suggest that regulation of AQP2 trafficking by Clec4d and Clec4n represents a new molecular mechanism through which AQP2 and urinary concentration are modulated.

Similar to NEKO mice, dioxin induces hydronephrosis without anatomical obstruction in the ureter[31]. The dioxin-induced hydronephrosis is elicited by elevated production of prostaglandin E2 (PGE$_2$)[31], which is an electrophilic inducer of Nrf2 (ref. 32). It is interesting to note that the development of hydronephrosis is induced by exposure to dioxin in the neonatal stage but not during adulthood[33]. This developmental stage-specific effect of dioxin (or window to dioxin) appears to be similar to our observation that Keap1-TKO mice develop hydronephrosis after DOX treatment during development but not in adulthood. These similarities imply that dioxin-induced hydronephrosis may be explained by the constitutive activation of Nrf2 by chronic elevation of PGE$_2$ in the developing kidney. This suggests that chronic exposure to Nrf2-activating chemicals, including environmental pollutants, prostaglandins and oxidative stresses, during kidney development may cause NDI and hydronephrosis.

It is interesting to note that Bartter's syndrome, another model for polyuria, is caused by rare inherited defects in the thick ascending limb of Henle's loop[34]. We found in this study that the diluting function of the ascending limb of Henle's loop is intact in NEKO mice, indicating that polyuria in NEKO mice is distinct from that of Bartter's syndrome. Nonetheless, inhibiting $PGE_2$ with indomethacin is clinically useful and widely used for Bartter's syndrome[35]. This suggests that $PGE_2$ production might contribute to the development of polyuria via dual mechanisms that impair the function of the counter current system in the ascending limb of Henle's loop, and the AVP–AQP2 system in the collecting ducts. In addition, it has been reported that mutations in Kelch-like 3 (KLHL3), a member of BTB domain containing Kelch protein, and Cul3 cause hypertension and electrolyte abnormalities[36,37], suggesting that Cul3 and members of KLHL family, such as Keap1 and KLHL3, play important roles in the maintenance of kidney homeostasis.

In light of the use of Nrf2-activating compounds as potential medical treatments, it has been shown that the timing at which the treatment is given needs to be carefully considered. In contrast, our results clearly suggest that the administration of Nrf2-activating chemicals during adulthood will not cause such adverse effects in the kidney, even if the treatment includes chronic administration of potent Nrf2 inducers. Indeed, many studies have shown that pharmacological Nrf2 activation gives rise to a protective effect against a variety of stresses, including kidney injury[38,39]. Bardoxolone methyl (CDDO-Me), a potent Nrf2 activator, has been studied in clinical trials, and the phase III trial of CDDO-Me was terminated due to a higher rate of heart failure events in advanced chronic kidney disease patients[40]. Many of these events appear to be due to fluid retention[41], although CDDO-Me-treated patients had a clinically meaningful increase of glomerular filtration rate[42]. In addition, treatment of Zucker diabetic fatty rats with RTA405, a synthetic triterpenoid analog of CDDO-Me, resulted in a reduction in urine volume[43]. These observations indicate that the adverse effects of CDDO-Me, namely, the fluid retention, are opposite to the phenotype observed in NEKO mice, suggesting that the adverse effects of CDDO-Me are pathology specific or independent of Nrf2 activation.

In addition to the renal phenotypes, we found mild anaemia, growth retardation and poor survival in NEKO mice. Anaemia, growth retardation and poor survival are not recapitulated in renal tubular-specific Keap1-deficient mice (Keap1-TKO), indicating that these phenotypes are independent of Nrf2 activation in the renal tubules. As loss of Keap1 in hematopoietic cells has been shown to suppress differentiation towards the erythroid lineage[10], the anaemia in NEKO mice is likely due to a loss of Keap1 in hematopoietic cells. The growth retardation of NEKO mice may be due to Nrf2 activation in the skeletal muscle and adipose tissues, as Keap1 deletion in the skeletal muscle reduces body weight[44], and loss of Keap1 represses differentiation of adipose cells[45]. As NEKO mice likely have other phenotypes that have not yet been studied, further analysis of NEKO mice will provide new insights to understand the physiological function of Nrf2.

In summary, we have generated NEKO mice harbouring constitutive Nrf2 activation throughout the whole body, without juvenile lethality. Through the analysis of NEKO mice, we identified a novel phenotype in which Nrf2 hyperactivation mediated by the loss of Keap1 in the developing kidney causes NDI and hydronephrosis. Our findings imply that Nrf2 activity should be controlled at an appropriate level during development in order to maintain renal homeostasis. Thus NEKO mice serve as a valuable experimental animal model with which to better understand our cytoprotective defense systems.

## Methods

**Mice.** Oesophageal Nrf2-deficient and systemic Keap1-null mice (NEKO, $Nrf2^{Flox/Flox}::Keratin5-Cre::Keap1^{-/-}$) were generated by crossbreeding $Nrf2^{Flox/Flox}$ (ref. 46) with K5-Cre[13] and $Keap1^{+/-}$ (ref. 8), while littermate mice ($Nrf2^{Flox/Flox}::K5-Cre:Keap1^{+/-}$ or $Nrf2^{Flox/Flox}::Keap1^{+/-}$) were used as controls. Neuron-specific Keap1 knockout mice (Keap1-NKO, $Keap1^{Flox/Flox}::Nestin-Cre$) were generated by crossbreeding $Keap1^{Flox/Flox}$ (ref. 11) with Nestin-Cre[47], while littermate mice ($Keap1^{Flox/Flox}$) were used as controls. Keap1 knockdown mice (Keap1-KD, $Keap1^{Flox/-}$) were previously generated[9], while littermate mice ($Keap1^{Flox/+}$) were used as controls. Keap1-Nrf2 double knockout mice ($Keap1^{-/-}::Nrf2^{-/-}$) were previously generated[8], while littermate mice ($Keap1^{+/-}::Nrf2^{-/-}$) were used as controls. Renal tubule-specific Keap1 knockout mice (Keap1-TKO, $Keap1^{Flox/Flox}::Pax8-rtTA::TetO-Cre$) were generated by crossbreeding $Keap1^{Flox/Flox}$ (ref. 12) with Pax8-rtTA[48] and TetO-Cre[49], while littermate mice ($Keap1^{Flox/Flox}$ or $Keap1^{Flox/Flox}::Pax8-rtTA$) were used as controls. Four-week-old mice or pregnant female mice were fed with $1 mg l^{-1}$ DOX in the drinking water[43]. After continuous DOX feeding, 8-week-old mice were subjected to analysis. All compound mutant mice examined in this study were from a mixed genetic background, with contributions from 129Sv/J and C57BL/6J strains. An almost equal combination of male and female mice was used in our experiments. The 24-h urine volume, drinking water volume and food consumption were measured using a metabolic cage, and osmolality was measured by SRL, Inc. Plasma AVP was quantified with a competitive enzyme-linked immunoassay ($Arg^8$-Vasopressin EIA Kit, Enzo Life Sciences). Blood samples were collected from the mice and analysed using the iSTAT-1 analyser (Abbott). Plasma blood urea nitrogen and creatinine were measured using FDCV7000V (Fujifilm). For the AVP stimulation test, mice were given water containing 5% sucrose before the experiment to make sure of production of dilute urine. Urine was collected and $1 ng g^{-1}$ body weight of (desamino-Cys$^1$, D-Arg$^8$) AVP (dDAVP, Sigma) was injected intraperitoneally. Urine osmolality was measured at 1 and 2 h after the injection. All mice were treated according to the regulations of The Standards for Human Care and Use of Laboratory Animals of Tohoku University (Sendai, Japan) and the Guidelines for Proper Conduct of Animal Experiments of the Ministry of Education, Culture, Sports, Science, and Technology of Japan. All animal in the experiments were killed with the approval of the Tohoku University Animal Care Committee.

**Gene expression analysis.** Total RNA was prepared from tissues and cells using a Sepazol-RNA I Super G RNA Extraction Kit (Nakalai). The cDNAs were synthesized from the total RNA using ReverTra Ace qPCR RT master mix with gRNA Remover (Toyobo). Real-time quantitative PCR was performed using StepOne or QuantStudio (Life Technologies). Primer and probe sequences are listed in Supplementary Table 1.

**Western blotting.** Kidney tissues were homogenized in 0.25 M sucrose. The homogenates were analysed by western blotting using anti-AQP2 (sc-9882; Santa Cruz; 1:200 dilution), SLC34A1 (Novus Biologicals; NBP2-13328; 1:1,000 dilution) and α-Tubulin (T9026, Sigma; 1:1,000 dilution) antibodies. Each urine sample was loaded with the same amount of creatinine. Full blottings are provided in Supplementary Fig. 15.

**Histological analysis.** Oesophagus and kidney from the mice were fixed with Mildform 10N and embedded in paraffin. Samples were subjected for periodic acid-Schiff staining or haematoxylin and eosin staining. For immunostaining, samples were stained using anti-AQP2 (C-17, sc-9882; Santa Cruz; 1:500 dilution), AQP4 (Millipore; AB3594; 1:400 dilution), NCC (Millipore; AB3553; 1:1,000 dilution) and anti-NQO1 (ab2346, Abcam; 1:1,000 dilution), and positive reactivity was visualized with diaminobenzidine staining.

**Renal collecting duct sorting by biotinylated DBA/streptavidin magnet beads.** After removing the kidney capsule, the kidney was immersed in 1 ml fresh-made digestion buffer ($0.52 U ml^{-1}$ liberase TM (Roche) and $3 U ml^{-1}$ DNase I (Roche) in Hanks balanced salt solution (Invitrogen)). Kidney was minced using scissors and incubated in a 37 °C water bath for 2 h, with gentle pipetting every 30 min. After incubation, the cells were filtrated through a 40-μm cell strainer (BD Bioscience) to remove clumps. For DBA magnetic sorting, the kidney cell suspension was incubated with biotinylated DBA (Vector Laboratories) in a cold room for 1 h. DBA-labelled cells were incubated with Dynabeads M-280 streptavidin (Life Technologies) and separated by magnet concentrator four times.

**ChIP-qPCR assay.** To obtain BMDMs, haemolysed bone marrow cells from wild-type and $Nrf2^{-/-}$ mice were cultured for 7 days in DMEM supplemented with 10% FBS, $20 ng ml^{-1}$ macrophage colony-stimulating factor (M-CSF, PeproTech) and antibiotic–antimycotic reagent (Life Technologies). At day 3, a half volume of fresh DMEM with 10% FBS and $20 ng ml^{-1}$ M-CSF was added. At day 7, BMDMs were depleted with M-CSF several hours before stimulation. BMDMs were treated with or without 100 μM DEM for 4 h and subjected to a ChIP assay. Succinctly, BMDMs were fixed with 1% formaldehyde and subsequently quenched with

glycine. After washing with PBS, fixed samples were suspended in cell lysis buffer (5 mM PIPES-KCl (pH 8.0), 85 mM KCl, and 0.5% NP40), and nuclei were collected and stored at 80 °C as nuclear pellets, or nuclear lysates were prepared by dissolving nuclei in a nucleus lysis buffer (50 mM Tris-HCl (pH 8.0), 10 mM EDTA and 1% SDS).

For Nrf2 ChIP analysis, lysates were thawed and sonicated with Sonifer (BRANSON) to obtain chromatin fragments of 300–1,000 bp, and the lysates were then diluted 10-fold with ChIP dilution buffer (16.7 mM Tris-HCl (pH 8.0), 1.2 mM EDTA, 0.01% SDS, 1.1% TrironX100 and 167 mM NaCl). After doubling dilution with a sonication buffer (90 mM HEPES (pH 7.9), 220 mM NaCl, 10 mM EDTA, 1% NP-40, 0.2% sodium deoxycholate and 0.2% SDS), nuclei were homogenized with a Bioruptor (Tosho-Denki). After washing, a complex of nuclear proteins, DNA and antibodies against Nrf2 (Cell Signaling technology; D1Z9C; 1:250 dilution) was retrieved with Protein A- and Protein G-Dynabeads (Life Technologies). After the crosslinking was reversed, chromatin fragments were treated with RNase A and proteinase K. DNA was purified with phenol–chloroform extraction or Ampure XP (Beckman 566 Coulter). In the ChIP-qPCR analyses, the values from the immunoprecipitated samples were normalized to that from the input DNA. Primer sequences are listed in Supplementary Table 1. ChIP-seq data were obtained from a previous study[24].

**Statistical analysis.** Values, including those displayed in the graphs, represent means ± s.e. Statistical analyses were performed using the unpaired *t*-test (Welch's *t*-test). Variation estimated by *F*-test was considered in *t*-test.

**Data availability statement.** Microarray data are deposited in GEO (Gene Expression Omnibus) under accession number GSE84780. Other data that support the findings of this study are available from the corresponding author upon reasonable request.

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

## Acknowledgements

We thank Dr Takashi Moriguchi, Dr Norio Suzuki, Dr Yu Lei and Dr Masahiro Nezu for discussion and the Biomedical Research Core of Tohoku University Graduate School of Medicine for technical support. This work was supported in part by grants from JSPS KAKENHI 26460354 and 26111010 to T.S., 26111002 to M.Y., AMED-CREST (chronic inflammation) to M.Y. and Mitsubishi Foundation and Takeda Science Foundation (to M.Y.).

## Author contributions

T.S., N.T., H.S. and M.Y. designed the research and analysed the data. T.S., S.S., K.H., E.N., E.H.K. and A.Y. conducted the experiments. T.S., L.B., N.T. and M.Y. wrote the paper.

## Additional information

**Competing financial interests:** The authors declare no competing financial interests.

