## [Peer Review File · Nature Communications]

Reviewers' comments:

Reviewer #1 (Remarks to the Author):

This study builds on the authors' previous reports that Keap1 deletion leads to hyper activation of the transcription factor Nrf2. In this study, the authors have generated a new NEKO mouse model (Keap1 deletion and high Nrf2 activity), obviating the juvenile lethality and hyperkeratosis associated with whole body Keap1 deletion. The NEKO mouse is characterised with polyuria and low osmolality and subsequent renal structural damage, resulting from a reduction in AQP2 protein levels.

The authors further demonstrate that dysregulation of AQP2 may be dependent on up-regulation of Clec4d and Clec4n genes during renal development. These conclusions were based on additional experimental data obtained from Keap1-TKO mice, in which Keap1 deletion induced at an embryonic stage (and not adult 8 weeks stage) exhibited polyuria. This finding suggested that the 'timing of Keap1 deletion' is critical for Nrf2 induced polyuria and dysregulation of AQP2.

The study is novel, experiments well described and the data support the authors' conclusions. Nevertheless, I have several specific points that I would ask the authors' to address.

Specific points:

1. p3, lines 62-3 - please reword this sentence...perhaps 'Cellular Nrf2 activation results from a depression mechanism involving decreased proteasomal degradation of the Keap1-Nrf2 complex.'
2. p6, lines 103-4 - what underlies the growth retardation in NEKO mice? The authors suggest that "Nrf2 activation in tissues other than squamous epithelium" underlie poor growth and survival of NEKO mice. Are similar findings observed in the Keap1-TKO mice following neonatal activation of Nrf2? Thus brief further discussion of the importance of neonatal hyperactivation of Nrf2 seems warranted.
3. p6, lines 113-14 - could further insight be provided as to why Keap1 null mice do not exhibit polyuria. Is there something unique occurring in the NEKO mouse in which hyperkeratosis is absent?
4. p7, lines 137-38 - Could some further insight be provided on the underlying post-

translational modifications of AQP2?

5. p10, lines 191-97 - Since administration of DOX to pregnant dams (but not adults) leads to polyuria, could the authors further discuss the importance of the 'developmental window' on subsequent renal dysfunction.

6. p11, lines 226032 - Some further insight into phenotypic differences in the NEKO and Keap1-TKO mice would seem warranted. Are there any concerns about the mouse background.

Reviewer #2 (Remarks to the Author):

Susuki et al. are demonstrating here that Nrf2 activation during development, but not during adulthood, leads to nephrogenic diabetes insipidus with dysregulation of AQP2 trafficking, probably induced by upregulation of Clca4d and Clca4n, gene members of the C-type lectin family.

The nephrogenic diabetes insipidus is demonstrated here with measurements of urine osmolalities and demonstration of progressive dilation of the urinary tract, as also observed in humans bearing AQP2 or AVPR2 mutations (Bockenhauer D, Bichet DG. Pathophysiology, diagnosis and management of nephrogenic diabetes insipidus. Nat Rev Nephrol. 2015 Oct;11(10):576-88).

Two important measures are missing: plasma sodium levels during dehydration with concomitant measurements of urine osmolality, and urine osmolality response to dDAVP. These are easy additions to the paper.

In the discussion there is an interesting report of dioxin-induced hydronephrosis with elevation of PGE₂, an electrophilic inducer of Nrf2. Not discussed by the authors is another model of polyuria and polydipsia, Bartter's syndrome (Uncompensated polyuria in a mouse model of Bartter's syndrome. Takahashi N, Chernavsky DR, Gomez RA, Igarashi P, Gitelman HJ, Smithies O. Proc Natl Acad Sci U S A. 2000 May 9;97(10):5434-9), where inhibiting PGE₂ with indomethacin is clinically useful and widely used (Neonates with Bartter syndrome have enormous fluid and sodium requirements. Azzi A, Chehade H, Deschênes G. Acta Paediatr. 2015 Jul;104(7):e294-9).

Finally the authors may comment on mutations in kelch-like3 and cullin 3 causing hypertension and electrolyte abnormalities as demonstrated by Rick Lifton and his team (Nature. 2012 Jan 22;482(7383):98-102 ; Proc Natl Acad Sci U S A. 2013 May 7;110(19):7838-43).

Reviewer #3 (Remarks to the Author):

In this study Suzuki et al describe the generation of a mouse model which uncovers a novel function for the Nrf2 transcription factor in the kidney. To overcome the perinatal lethality of Keap1^{-/-} mice the authors intercrossed this line with a Keratin5-Cre:Nrf2^{flox/flox} line. Using this clever strategy, they rescue the lethality of the mice due to the hyperkeratosis of the upper digestive tract. The mice survive for a longer period of time (up to 400 days) allowing to uncover a renal function for Nrf2 in regulation of urine concentration and aquaporin2 expression/maturation/excretion. To demonstrate that the effects are indeed due to a renal specific function of Nrf2 downstream of Keap1, the authors further generate a kidney-specific inducible Keap1^{flox/flox} line which recapitulates some of the features of the NEKO mice when doxycycline is administered during the embryonic stages (but not when administered to adults).

The study is potentially interesting and reports some intriguing observations. However, it is very descriptive, in some instances multiple controls are lacking and no mechanistic insight is provided. In particular, while the NEKO mice survive due to the esophageal rescue, Keap1 remains inactivated in every tissue and so the constitutive activation of the Nrf2 transcription factor. Thus, some of the observations made might be due to secondary effects driven by changes occurring elsewhere than the kidney. A second major concern is that the authors do not characterize the kidney phenotype, Nrf2 activity and AQP2 expression in a sufficiently accurate manner. Analysis is performed generically in "renal tubules" while markers for specific tubular segments should be employed. Finally, the downregulation of AQP2 and its secretion into the urine is poorly characterized at the molecular level and no credible mechanism is provided to explain these variations.

Specific comments:

- The title is not justified. The authors did not test any parameter for nephrogenic diabetes insipidus, they only observed polyuria. The data in the manuscript show that constitutive (almost ubiquitous) upregulation of Nrf2 causes downregulation of AQP2 in the kidney and polyuria.
- In figure 1 the authors show that the Nrf2 expression is decreased in the esophagus and also in the skin but not in the liver, lung, heart and kidney as expected. What is the phenotype in these other organs and could this indirectly influence the renal phenotype (and viceversa)? What is the cause of death of the NEKO mice? And what is the reason for decreased gain in body weight?
- Figure 2 and 3 are interesting, but I am not sure the authors are interpreting the data correctly. They find polyuria, reduced osmolality, increased water intake, increased mRNA levels of AVP in the hypothalamus and increased circulating AVP, in the presence of a downregulated AQP2 in the kidney. It appears that because the mice lack AQP2 they fail to

reabsorbe water, explaining the polyuria and decreased osmolality. The increased AVP is probably an attempt to rescue the problem. I would expect that the vasopressin receptor activity and the entire cascade downstream in the kidney is upregulated. Despite this, having a very reduced level of AQ2 the signaling cascade cannot appropriately expose sufficient AQ2 on the apical side. So, increased AVP is not surprising at all, but actually expected. The authors should look into the signaling cascade of vasopressin receptor in detail in these kidneys. Alternatively, the excessive stimulation of the vasopressin receptor by VPA might result in the long-term exhaustion of the AQ2 pool or else in secretion via exosomes. All these possibilities should be looked at carefully. What is it seen first? Upregulation of the AVP transcript in the hypothalamus or decreased AQ2 in the kidney?

- Figure 3d: a marker of collecting ducts should be used as a loading control here. The histology in g and h shows that the NEKO kidneys have morphological defects. The AQ2-positive tubules are clearly reduced and morphologically different from the control. Thus, one possibility is that inactivation of Keap1 and upregulation of Nrf2 causes defective development of the tubular segments. This would be in line with the Dox inducible mice showing defects only when Keap1 is inactivated during development. Staining of markers for the different segments of the renal tubule should be included for the stainings in general throughout the paper.
- Figure 5: the NQO1 staining in the Keap1^{-/-} kidneys should be performed by counterstaining with different tubular markers (proximal, distal and collecting ducts).
- Figure 7: the Keap TKO mice are also problematic because the Cre line employed is active throughout the renal tubule (rtTAPax8Cre). A more distal/collecting duct Cre line would have been desirable. The phenotype in the Keap1-TKO in b is very different from the phenotype of the NEKO, which is surprising.
- Figure 8: urinary excretion of AQ2 is potentially interesting, but an observation that would require per se a fine characterization. It was previously described that AQ2 could be present in exosomes. Were exosomes isolated from the urine?
- The authors performed microarrays analysis, the complete list of genes identified and fold changes should be deposited in GEO and should be shown in the supplementary material.
- The two genes that they found changing most are clec4n and clec4d, proteins of unknown function containing lectin domains. Based on the presence of these lectin domains they make a wild link with a potential role of galectin in protein trafficking (ref 21). Thus, the study does not contain any data to show or potentially explain a connection between NRF2 upregulation and aquaporin downregulation, an observation that remains potentially interesting.

Reviewer #4 (Remarks to the Author):

Suzuki et al. used an elegant mouse genetics strategy to uncover a novel role of the Nrf2

transcription factor in early tubular development and in the pathogenesis of nephrogenic diabetes insipidus in mice. The results further suggest that a reduction in the levels of aquaporin 2 is responsible for the phenotype. The phenotype is unexpected and nicely described, but there is a lack of mechanistic insight. First, it is unclear if *Clec4d* and *Clec4n* are direct targets of Nrf2. Second, it is not clear if their abnormal expression indeed affects aquaporin 2 trafficking. Finally, the role of aquaporin 2 in the phenotype remains speculative.

Specific points:

1. There are different types of K5-Cre mice, which target the esophagus to a different extent. Indeed, Fig.1a suggests that there is only a mild reduction in *Nqo1* expression in the esophagus of NEKO mice, whereas the reduction in the skin is much more pronounced. Therefore, it is critical to understand to what extent the esophagus phenotype is rescued in NEKO mice. A mild esophagus phenotype may also contribute to the limited life expectancy of the mice. The thickness of the esophageal stratum corneum should be quantified in the upper, middle and lower esophagus. This should be correlated with Nrf2 and *Nqo1* expression in the different parts of the esophagus.
- 2.) Fig. 1a: The authors should analyze at least one more Nrf2 target gene, to get a better idea about the level of Nrf2 activation in the different mouse models.
- 3.) Fig1b-e/f. The authors previously also showed hyperkeratosis in the forestomach of *Keap1* ko mice (Wakabayashi et al., 2001). Is this phenotype also rescued in the NEKO mice?
- 4.) Can the authors really exclude malnutrition as a cause of the lack of weight gain? Measurement of serum glucose levels would be helpful to further test this possibility.
- 5.) Fig. 3d: A negative control should be shown from another tissue (or antibody pre-incubated with the immunization peptide) to determine if the lower band is indeed non-glycosylated aquaporin -2. In addition, molecular weights in kDa should be indicated.
- 6.) Aquaporin 2 is regulated by alternative splicing (Suzuki et al., 2016). Do the primers used for the detection of aquaporin-2 mRNA detect all splice variants?
- 7.) It is shown that early deletion of *Keap1* during embryogenesis is responsible for the kidney phenotype, whereas deletion in the adult animal had no effect. It is important to determine if aquaporin 2 proteins levels are also affected upon deletion of *Keap1* in adult mice. In this case it is unlikely that the defect is due to the reduced levels of this protein.
- 8.) Fig. 4h,i: It would be interesting to know when the inflammation occurs to determine

whether it is a cause or consequence of the hydronephrosis. The inflammation should be further characterized, e.g. by flow cytometry analysis of different immune cells.

9.) Fig. 8a lacks a loading control, e.g. Ponceau S staining of the membrane. The molecular weight should be indicated.

10.) Where are Clec4d and Clec4n expressed and what is the expression pattern in the different Keap1ko lines? In addition, it would be nice to see protein data.

11.) It remains unclear whether Clec4d and Clec4n are direct Nrf2 targets. This is an important issue that should be experimentally addressed, e.g. by CHIP.

12.) The paper lacks functional studies (e.g. in cultured cells) showing that Clec4d/n upregulation indeed affects aquaporin-2 glycosylation/trafficking.

13.) Finally, it remains unclear if the Clec4d/n - aquaporin 2 axis is indeed responsible for the observed hydronephritis and polyuria in Keap1 TKO and NKO mice. This requires a phenocopy or rescue experiment.

14.) Please indicate N numbers in the legends to Fig. 3b and f, 6b, 6d, 7f, 8d and e.

To Reviewer #1:

The study is novel, experiments well described and the data support the authors' conclusions. Nevertheless, I have several specific points that I would ask the authors' to address.

We appreciate the reviewer for the constructive and helpful comments and advices.

Specific points:

1. p3, lines 62-3 - please reword this sentence...perhaps 'Cellular Nrf2 activation results from a depression mechanism involving decreased proteasomal degradation of the Keap1-Nrf2 complex.'

We thank the reviewer for this professional comment. We have improved the sentence as suggested.

2. p6, lines 103-4 - what underlies the growth retardation in NEKO mice? The authors suggest that "Nrf2 activation in tissues other than squamous epithelium" underlie poor growth and survival of NEKO mice. Are similar findings observed in the Keap1-TKO mice following neonatal activation of Nrf2? Thus brief further discussion of the importance of neonatal hyperactivation of Nrf2 seems warranted.

We thank the reviewer for this important comment. We have examined the growth and survival of the Keap1-TKO mice following neonatal activation of Nrf2. Our new data indicate that the Keap1-TKO mice do not show growth retardation upon the neonatal activation of Nrf2, suggesting that the poor growth and survival of NEKO mice is due to Nrf2 activation in the other tissues than renal tubules. We have added these new data (Supplementary fig. 11) to the revised manuscript. Also, following the advice by the reviewer, we have discussed the importance of neonatal hyperactivation of Nrf2 (p18, lines 380-382)

3. p6, lines 113-14 - could further insight be provided as to why Keap1 null mice do not exhibit polyuria. Is there something unique occurring in the NEKO mouse in which hyperkeratosis is absent?

We apologize for the lack of clear explanation. Keap1-null mice are lethal within three weeks after birth (Nature Genetics, 2003) and measurement of urine volume is impossible. Therefore, we have been utilizing Keap1 “knockdown” mice (MCB, 2010), which show mild activation of Nrf2 in whole body due to the reduction of Keap1 expression. The Keap1 knockdown mice do not exhibit polyuria. In contrast, we found in this study that NEKO mice show marked polyuria. As NEKO mice have been developed to avoid the neonatal lethality, and the mice display complete loss-of-Keap1 activity and full-activation of Nrf2 except for the squamous epithelial tissues, we surmise that the polyuria phenotype requires the full-activation of Nrf2.

4. p7, lines 137-38 - Could some further insight be provided on the underlying post-translational modifications of AQP2?

We thank the reviewer for this professional comment. Indeed, we found that glycosylation form of AQP2 is significantly decreased in the kidney of NEKO mice. This result suggests that apical membrane trafficking and shedding of AQP2 may be

stimulated in NEKO mouse kidney. We have added a sentence explaining this result to the revised manuscript (p8, lines 142-143).

5. p10, lines 191-97 - *Since administration of DOX to pregnant dams (but not adults) leads to polyuria, could the authors further discuss the importance of the 'developmental window' on subsequent renal dysfunction.*

We appreciate the reviewer for this comment. We added a succinct discussion regarding the developmental window of Nrf2 activation to the revised manuscript (p11, lines 234-235)

6. p11, lines 226-32 - *Some further insight into phenotypic differences in the NEKO and Keap1-TKO mice would seem warranted. Are there any concerns about the mouse background.*

We apologize for the lack of clarity in our explanation. While NEKO mice lack Keap1 expression in whole body (including kidney tubules) except squamous epithelia, Keap1-TKO mice lacks the Keap1 expression only in kidney tubules. Therefore, we believe that the phenotypic differences between NEKO mice and Keap1-TKO mice should be elicited by the lack of Keap1 in the other tissues. As for the mouse background, all compound mutant mice examined in this study are from mixed genetic backgrounds, with contributions from C57BL/6J and 129Sv/J strains. We surmise that the mouse background does not affect much for the phenotypic differences of the NEKO mice and Keap1-TKO mice.

To Reviewer #2:

Susuki et al. are demonstrating here that Nrf2 activation during development, but not during adulthood, leads to nephrogenic diabetes insipidus with dysregulation of AQP2 trafficking, probably induced by upregulation of Clec4d and Clec4n, gene members of the C-type lectin family.

We thank the reviewer for the constructive comments.

Two important measures are missing: plasma sodium levels during dehydration with concomitant measurements of urine osmolality, and urine osmolality response to dDAVP. These are easy addition to the paper.

We thank the reviewer for this important comment and we are happy to report the results of our experiments. We have examined plasma sodium levels during dehydration with concomitant measurements of urine osmolality. The results demonstrate that plasma levels of sodium, potassium and chloride in NEKO mice are increased during dehydration, while those of control mice are unchanged (Figure 5a-c). In addition, NEKO mice display urinary concentration defect during dehydration (Figure 5d). These results thus indicate that NEKO mice cannot make concentrated urine in response to dehydration. To examine NEKO mouse response to dDAVP, we also measured urine osmolality of NEKO mice before and after dDAVP injection. NEKO mice cannot increase the urine osmolality in response to dDAVP administration (Figure 5e). These results clearly indicate the presence of nephrogenic diabetes insipidus in NEKO mice. We have added these results into the revised manuscript (p10, lines 191-201).

In the discussion there is an interesting report of dioxin-induced hydronephrosis with elevation of PGE2, an electrophilic inducer of Nrf2. Not discussed by the authors is another model of polyuria and polydipsia, Bartter's syndrome (Uncompensated polyuria in a mouse model of Bartter's syndrome. Takahashi N, Chernavvsky DR, Gomez RA, Igarashi P, Gitelman HJ, Smithies O. Proc Natl Acad Sci U S A. 2000 May 9;97(10):5434-9), where inhibiting PGE2 with indomethacin is clinically useful and widely used (Neonates with Bartter syndrome have enormous fluid and sodium requirements. Azzi A, Chehade H, DeschActa Paediatr. 2015 Jul;104(7):e294-9).

We thank the reviewer for this professional comment. As the reviewer pointed out, inhibition of PGE2, an electrophilic inducer of Nrf2, with indomethacin is clinically useful for the treatment of Bartter's syndrome. This observation is consistent with the notion that PGE2-mediated Nrf2 activation may be involved in development of polyuria.

As TAL is impaired in Bartter's syndrome, we have examined the TAL function of NEKO mice. To this end, we have measured urine osmolality of NEKO mice after water loading. The water challenge significantly decreased urine osmolality to the levels below that of serum (Supplementary fig. 7), indicating that diluting function of TAL is not compromised in NEKO mice. These observations indicate that polyuria of NEKO mice is due to a defect of water reabsorption in the collecting duct, not due to a diluting defect of TAL. We have added these results and related discussion to the revised manuscript (p10, lines 202-210).

Finally the authors may comment on mutations in kelch-like3 and cullin 3 causing hypertension and electrolyte abnormalities as demonstrated by Rick Lifton and his team

(Nature. 2012 Jan 22;482(7383):98-102 ; Proc Natl Acad Sci U S A. 2013 May 7;110(19):7838-43).

We thank the reviewer for this professional comment. As advised by the reviewer, we have added a succinct discussion regarding mutations in Kelch-like3 (KLHL3) and Cullin 3 (Cul3) to the revised manuscript. We surmise that Cul3 and KLHL family members such as Keap1 and KLHL3 play important roles for kidney homeostasis (p18, lines 375-379).

To Reviewer #3:

The study is potentially interesting and reports some intriguing observations. However, it is very descriptive, in some instances multiple controls are lacking and no mechanistic insight is provided. In particular, while the NEKO mice survive due to the esophageal rescue, Keap1 remains inactivated in every tissue and so the constitutive activation of the Nrf2 transcription factor. Thus, some of the observations made might be due to secondary effects driven by changes occurring elsewhere than the kidney.

We thank the reviewer for helping us by reviewing our manuscript. As for the concern related to the secondary effects, we believe that our results from renal tubular specific Keap1 knockout (Keap1-TKO) mice clearly exclude the possibility that “secondary effects by changes occurring elsewhere than in the kidney”, but demonstrate that polyuria is due to Nrf2 activation in kidney.

A second major concern is that the authors do not characterize the kidney phenotype, Nrf2 activity and AQP2 expression in a sufficiently accurate manner. Analysis is performed generically in "renal tubules" while markers for specific tubular segments should be employed.

Our search of the literatures indicates that Pax8-rtTA is a well-established transgenic mouse line that expresses the reverse tetracycline-dependent transactivator (rtTA) to proximal and distal tubules plus entire collecting duct system of kidney (Traykova-Brauch et al, *Nature Medicine* 2008 and many more). We have confirmed the observation by ourselves utilizing Rosa26TdTomato mice (Yu et al, *MCB* 2014). Therefore, we believe that *Keap1* gene must be deleted in practically all the renal tubules and “renal tubules” is an appropriate term here. We agree that narrowing down the responsible tubular segments for finer mechanistic study is important, but we also feel it is beyond the scope of this paper.

Finally, the downregulation of AQP2 and its secretion into the urine is poorly characterized at the molecular level and no credible mechanism is provided to explain these variations.

We found in this revision that Clec4d and Clec4n are in fact expressed in sorted collecting duct cells (**Supplementary fig. 13**). This new finding further supports the notion that Clec4d and Clec4n expressions in the collecting duct cells enhance the AQP2 trafficking.

To further test the hypothesis that Clec4d and Clec4n bind to glycosylated form of AQP2 and enhance excretion of AQP2, we need a cell culture experiment system. To our best knowledge, however, there is no collecting duct cell line expressing AQP2. Therefore, in this revision we attempted to generate a stable cell line expressing AQP2 by using M-1 cell line, a collecting duct-derived cell line. To our disappointment, we could not find glycosylated form of AQP2 in the established stable cell line (**Supplementary figure for reviewer only 1**). This is a quite different situation observed in animals. Thus, in the current situation it is technically not feasible to evaluate the influence of Clec4d and Clec4n on glycosylated form of AQP2 using a cell culture system. Given the distance we have been developing by using the NEKO study, we feel that fine mechanistic insights will be a issue that should be addressed in the future paper.

Specific comments:

- *The title is not justified. The authors did not test any parameter for nephrogenic diabetes insipidus, they only observed polyuria. The data in the manuscript show that constitutive (almost ubiquitous) upregulation of Nrf2 causes downregulation of AQP2 in the kidney and polyuria.*

We have added new results to the revised manuscript in which we have tested the presence of nephrogenic diabetes insipidus (NDI). The results clearly show that NEKO mice cannot increase urine osmolality in response to dehydration or dDAVP administration (Figure 5). These results strongly support the notion that NEKO mice suffer from NDI.

Our results from renal tubular-specific Keap1 knockout mice (Keap1-TKO) clearly demonstrate that polyuria is caused by Nrf2 activation in the kidney, but not due to “ubiquitous” Nrf2 activation in whole body or in the other tissues. In this sense, we think our title is justified.

- *In figure 1 the authors show that the Nrf2 expression is decreased in the esophagus and also in the skin but not in the liver, lung, heart and kidney as expected. What is the phenotype in these other organs and could this indirectly influence the renal phenotype (and vice versa)? What is the cause of death of the NEKO mice? And what is the reason for decreased gain in body weight?*

As answered above (to first general comment and to first specific comment), our results from renal tubular-specific Keap1 knockout (Keap1-TKO) mice clearly demonstrate that polyuria is due to Nrf2 activation in kidney, but not due to ubiquitous Nrf2 activation.

The poor survival and decreased gain in body weight of the NEKO mice are not recapitulated in Keap1-TKO mice, indicating that the cause of death of the NEKO mice is independent of the renal phenotype (Supplementary fig. 10 and 11).

- *Figure 2 and 3 are interesting, but I am not sure the authors are interpreting the data correctly. They find polyuria, reduced osmolality, increased water intake, increased mRNA levels of AVP in the hypothalamus and increased circulating AVP, in the presence of a downregulated AQP2 in the kidney. It appears that because the mice lack AQP2 they fail to reabsorb water, explaining the polyuria and decreased osmolality. The increased AVP is probably an attempt to rescue the problem. I would expect that the vasopressin receptor activity and the entire cascade downstream in the kidney is upregulated. Despite this, having a very reduced level of AQP2 the signaling cascade cannot appropriately expose sufficient AQP2 on the apical side. So, increased AVP is not surprising at all, but actually expected. The authors should look into the signaling cascade of vasopressin receptor in detail in these kidneys. Alternatively, the excessive stimulation of the vasopressin receptor by VPA might result in the long-term exhaustion of the AQP2 pool or else in secretion via exosomes. All these possibilities should be looked at carefully. What is it seen first? Upregulation of the AVP transcript in the hypothalamus or decreased AQP2 in the kidney?*

We would like to ask the reviewer to understand that polyuria is recapitulated in Keap1-TKO, demonstrating that polyuria is due to Nrf2 activation in the kidney but not due to neuronal Nrf2 activation. In addition, we found that reduction of AQP2 protein was observed even at 10 days of age in NEKO mouse kidney (Supplementary fig. 4). Therefore, we interpret that the increase of AVP is probably an attempt to rescue the loss

of water in NEKO mice.

- *Figure 3d: a marker of collecting ducts should be used as a loading control here. The histology in g and h shows that the NEKO kidneys have morphological defects. The AQP2-positive tubules are clearly reduced and morphologically different from the control. Thus, one possibility is that inactivation of Keap1 and upregulation of Nrf2 causes defective development of the tubular segments. This would be in line with the Dox inducible mice showing defects only when Keap1 is inactivated during development. Staining of markers for the different segments of the renal tubule should be included for the stainings in general throughout the paper.*

Although we tried to use AQP4 as a loading control for a marker of collecting duct cells, to our disappointment anti-AQP4 antibody did not work for Western blotting. In contrast, for immunohistochemistry, the anti-AQP4 antibody worked well to stain collecting duct. We found that the AQP4 staining was unchanged in Keap1-null mice compared to control mice (Supplementary fig. 5c and d), while AQP2 staining showed weaker signals in collecting ducts in Keap1-null mice (Supplementary fig. 5a and b). In addition to AQP2 reduction, morphological defect of collecting ducts was observed in Keap1-null mice at 10 days of age (Supplementary fig. 5a-d), implying that AQP2 reduction might cause morphological change of collecting ducts. It seems noteworthy that staining of NCC, a marker for distal convoluting tubule (DCT) was not changed in Keap1-null mice (Supplementary fig. 5e and f).

As the reviewer pointed out, we cannot exclude the possibility that inactivation of Keap1 and upregulation of Nrf2 causes defective development of the tubular segments. This point does not affect our current conclusion, but needs to be investigated in the future.

- *Figure 5: the NQO1 staining in the Keap1-/- kidneys should be performed by counterstaining with different tubular markers (proximal, distal and collecting ducts).*

Counterstaining with NQO1 and tubular markers is technically not feasible due to difficulty of the antibody combinations. Staining of NCC, a marker for distal convoluted tubule (DCT) was unchanged between control and NEKO mice (Supplementary fig. 5e and f). Although we have tried to stain several tubular markers, they did not work well except for the anti-NCC antibody.

- *Figure 7: the Keap TKO mice are also problematic because the Cre line employed is active throughout the renal tubule (rtTAPax8Cre). A more distal/collecting duct Cre line would have been desirable. The phenotype in the Keap1-TKO in b is very different from the phenotype of the NEKO, which is surprising.*

While further experiments with specific Cre lines are important for the understanding of the responsible tubules, it will certainly take time and efforts. The fact that anemia in NEKO mice was not recapitulated in the Keap1-TKO mice should be interpreted that anemia is independent of renal tubular phenotypes.

- *Figure 8: urinary excretion of AQP2 is potentially interesting, but an observation that would require per se a fine characterization. It was previously described that AQP2 could be present in exosomes. Were exosomes isolated from the urine?*

Indeed we preliminary tried to isolate exosomes from the urine, but we could not detect

AQP2 protein in the isolated urinary exosome.

- *The authors performed microarrays analysis, the complete list of genes identified and fold changes should be deposited in GEO and should be shown in the supplementary material.*

We have deposited the microarray data in GEO.

- The two genes that they found changing most are clec4n and clec4d, proteins of unknown function containing lectin domains. Based on the presence of these lectin domains they make a wild link with a potential role of galectin in protein trafficking (ref 21). Thus, the study does not contain any data to show or potentially explain a connection between NRF2 upregulation and aquaporin downregulation, an observation that remains potentially interesting.

We have present results that support the connection between Clec4d/Clec4n and Nrf2/Aqp2 in this paper. In this revision, we added a couple of new results. One is that Clec4d and Clec4n are indeed expressed *in vivo* in sorted collecting ducts (Supplementary fig. 13), and the other is that Nrf2 actually binds to Clec4d/Clec4n genes in ChIP seq and manual ChIP (Supplementary fig. 14). These results further support our contention that Clec4d and Clec4n in the collecting ducts enhance the AQP2 trafficking.

To Reviewer #4:

Suzuki et al. used an elegant mouse genetics strategy to uncover a novel role of the Nrf2 transcription factor in early tubular development and in the pathogenesis of nephrogenic diabetes insipidus in mice. The results further suggest that a reduction in the levels of aquaporin 2 is responsible for the phenotype. The phenotype is unexpected and nicely described, but there is a lack of mechanistic insight. First, it is unclear if Clec4d and Clec4n are direct targets of Nrf2. Second, it is not clear if their abnormal expression indeed affects aquaporin 2 trafficking. Finally, the role of aquaporin 2 in the phenotype remains speculative.

We thank the reviewer for valuable comments. To answer the reviewer's comments, we have conducted new experiments and added new data to the revised manuscript. Succinctly, we found that 1) Clec4d and Clec4n are indeed expressed in collecting ducts of NEKO mice, 2) Clec4d and Clec4n are direct target genes of Nrf2, and 3) a reduction of AQP2 is quite consistent with the polyuria phenotype.

Specific points:

1. There are different types of K5-Cre mice, which target the esophagus to a different extent. Indeed, Fig.1a suggests that there is only a mild reduction in Nqo1 expression in the esophagus of NEKO mice, whereas the reduction in the skin is much more pronounced. Therefore, it is critical to understand to what extent the esophagus phenotype is rescued in NEKO mice. A mild esophagus phenotype may also contribute to the limited life expectancy of the mice. The thickness of the esophageal stratum corneum should be quantified in the upper, middle and lower esophagus. This should be correlated with Nrf2 and Nqo1 expression in the different parts of the esophagus.

We thank the reviewer for this insightful comment. We surmise that the mild reduction in the *Nqo1* expression in the NEKO mouse esophagus is because the esophagus contains cells other than squamous epithelium, such as muscle. Indeed, food intake of NEKO mice is comparable, but water intake is much higher than those of control mice, respectively.

2.) Fig. 1a: The authors should analyze at least one more Nrf2 target gene, to get a better idea about the level of Nrf2 activation in the different mouse models.

We have examined expression of *Gclc*, which is another representative Nrf2 target gene. Expression pattern of *Gclc* was similar to that of *Nqo1*, indicating that the Nrf2 activity in NEKO mice is well verified. The result is added to **Supplementary fig. 1**.

3.) Fig1b-e/f. The authors previously also showed hyperkeratosis in the forestomach of Keap1 ko mice (Wakabayashi et al., 2001). Is this phenotype also rescued in the NEKO mice?

We thank the professional comment. The hyperkeratosis in the forestomach of Keap1-null mice is also rescued in the NEKO mice. We showed the phenotype of forestomach in **Supplementary fig. 2**.

4.) Can the authors really exclude malnutrition as a cause of the lack of weight gain? Measurement of serum glucose levels would be helpful to further test this possibility.

Blood glucose level in NEKO mice is comparable to that of control mice, supporting the

notion that there is no significant malnutrition in NEKO mice. We have added the blood glucose data to Supplementary fig. 3.

5.) *Fig. 3d: A negative control should be shown from another tissue (or antibody pre-incubated with the immunization peptide) to determine if the lower band is indeed non-glycosylated aquaporin -2. In addition, molecular weights in kDa should be indicated.*

Molecular weights in kDa have been indicated in the revised manuscript. To verify whether the lower band is non-glycosylated form of AQP2, we have transfected AQP2 expression vector into culture cells. Major band is approximately 29 kDa, which corresponds to non-glycosylated form of AQP2 (Nejsum et al, 2001) (Supplementary figure for reviewer only 1).

6.) *Aquaporin 2 is regulated by alternative splicing (Suzuki et al., 2016). Do the primers used for the detection of aquaporin-2 mRNA detect all splice variants?*

We thank the reviewer for this professional comment. The primers used in this study detect both authentic and alternative form of AQP2. To separately detect these forms of AQP2, we have designed specific primers for the authentic and alternative form of AQP2. However, while authentic form of AQP2 is detected, alternative form of AQP2 is not detected in kidneys of control and NEKO mice (Supplementary figure for reviewer only 2). We surmise this result implies that only authentic form of AQP2 is expressed in control and NEKO mice.

7.) *It is shown that early deletion of Keap1 during embryogenesis is responsible for the kidney phenotype, whereas deletion in the adult animal had no effect. It is important to determine if aquaporin 2 proteins levels are also affected upon deletion of Keap1 in adult mice. In this case it is unlikely that the defect is due to the reduced levels of this protein.*

We have examined AQP2 protein level in kidney of Keap1-TKO mice treated with DOX during adult stage and found that there is still remains sufficient amount of the AQP2 and glycosylated form of AQP2 proteins (Supplementary fig. 9), which is clear contrast to the case of NEKO mouse kidney. The results indicate that the urinary concentrating defect is likely due to reduction of AQP2.

8.) *Fig. 4h,i: It would be interesting to know when the inflammation occurs to determine whether it is a cause or consequence of the hydronephrosis. The inflammation should be further characterized, e.g. by flow cytometry analysis of different immune cells.*

Many studies have shown that Nrf2 plays roles to suppress inflammation. For instance, we recently published a paper related to this topic (Kobayashi et al, *Nature Commun* 2016). Therefore, we surmise that the inflammation in NEKO mice is unlikely a cause of the hydronephrosis, but rather a consequence of the hydronephrosis.

9.) *Fig. 8a lacks a loading control, e.g. Ponceau S staining of the membrane. The molecular weight should be indicated.*

We agree with this comment and molecular weights in kDa have been indicated in the revised manuscript. We also have added Ponceau S staining of the membrane as a loading control (Fig 8a). As loading amount was adjusted by creatine, protein amount of urine from NEKO mice is rather lower than that of control mice.

10.) *Where are Clec4d and Clec4n expressed and what is the expression pattern in the different Keap1ko lines? In addition, it would be nice to see protein data.*

We thank the reviewer for this professional comment. We have separated collecting duct cells from the kidney by biotinylated DBA/streptavidin magnetic beads, and examined expression levels of Clec4d and Clec4n. We found that expression of *Clec4d* and *Clec4n* genes in the collecting ducts is indeed induced in NEKO mice compared to control mice (Supplementary fig 13). We could not find good antibodies for Clec4d and Clec4n proteins, so that detection of these proteins could not be attained.

11.) *It remains unclear whether Clec4d and Clec4n are direct Nrf2 targets. This is an important issue that should be experimentally addressed, e.g. by ChIP.*

We agree with this comment and experimentally approached for this issue. We first exploited our recent analyses on anti-Nrf2 ChIP-sequencing analysis using bone marrow-derived macrophages (BMDMs) (Kobayashi et al, *Nature Commun* 2016). In this analysis, we found that Nrf2 binds to Clec4d and Clec4n gene loci and regulates their expression in BMDMs. We have validated this information that expressions of *Clec4d* and *Clec4n* genes in BMDMs are induced by treatment with DEM, an Nrf2 inducer, in an Nrf2-dependent manner (Supplementary fig. 14a and b). In addition, manual ChIP assays revealed that Nrf2 binds to *Clec4d* and *Clec4n* gene loci in BMDMs (Supplementary fig. 14c and d). These results strongly support the notion that *Clec4d* and *Clec4n* are direct target genes of Nrf2.

12.) *The paper lacks functional studies (e.g. in cultured cells) showing that Clec4d/n upregulation indeed affects aquaporin-2 glycosylation/trafficking.*

We would like to ask the reviewer to understand that functional studies using collecting duct culture cells are not well developed in the research society. To our best knowledge, there is no cell line expressing AQP2. Therefore, in order to further validate the hypothesis that Clec4n and Clec4d bind to glycosylated form of AQP2 and enhance excretion of glycosylated form of AQP2, we have attempted to establish a culture cell experimental system in this revision period. We have tried to generate a stable cell line expressing AQP2 by using M-1 cell line, a collecting duct derived cell line. To our disappointment, however, we could not find the glycosylated form of AQP2 (Supplementary figure for reviewer only 1). This is a quite different situation observed in animals. Thus, in the current situation it is technically not feasible to evaluate the influence of Clec4d and Clec4n on glycosylated form of AQP2 using a cell culture system. Given the distance we have been developing by using the NEKO study, we feel that fine mechanistic insights are issues that will be addressed in the future.

13.) *Finally, it remains unclear if the Clec4d/n - aquaporin 2 axis is indeed responsible for the observed hydronephritis and polyuria in Keap1 TKO and NKO mice. This requires a phenocopy or rescue experiment.*

We would like to ask the reviewer to understand that we have present results that support sufficiently the connection between Clec4d/Clec4n and Aqp2 in this paper. For instance, induction of Clec4d/Clec4n expression is observed in NEKO mice and Keap1-TKO mice treated with DOX from embryonic stage, but not in Keap1-TKO mice treated with DOX from adult stage (Fig. 8). The emergence of Clec4d/Clec4n induction is quite consistent

with those of hydronephrosis and polyuria.

In this revision, we also added multiple new results. We found that reduction of AQP2 protein is observed even at 10 days of age in NEKO mouse kidney (Supplementary fig. 4), where AQP2 accumulates in apical membrane of collecting ducts (Supplementary fig. 5). This observation strongly supports the notion that Clec4d and Clec4n in the collecting ducts enhance the AQP2 trafficking. In addition, we found that there remains sufficient amount of AQP2 and glycosylated form of AQP2 proteins in the kidney of Keap1-TKO mice treated with DOX during adult stage (Supplementary fig. 9), which is quite consistent with the weakened induction of Clec4d/Clec4n in the mouse kidney (Fig. 8). These observations further support our conclusion that AQP2 is responsible for development of hydronephrosis and polyuria. We feel that rescue experiments will be an issue that should be addressed in the future.

14.) Please indicate N numbers in the legends to Fig. 3b and f, 6b, 6d, 7f, 8d and e.

We have indicated N numbers in the legends.

Reviewers' comments:

Reviewer #1 (Remarks to the Author):

This Reviewer is satisfied with the Authors' response to the comments raised and finds the study novel and of interest to a wide field of research.

Reviewer #2 (Remarks to the Author):

My previous comments related to a better demonstration of the nephrogenic diabetes insipidus state have been addressed by the additional experiments now described in Fig 5 a-c. The supplemental fig 7 demonstrating a normal diluting function is also of interest.

Minor: line 197, page 10: dDAVP is a synthetic vasopressin analog, please add analog

Reviewer #3 (Remarks to the Author):

The authors have made attempts at answering to most of the questions raised. However, one major point raised by this reviewer remains non-satisfactorily addressed: the role of AVP levels. This reviewer understands extremely well the importance of confirming the effects observed in the NEKO mice using kidney-specific inactivation of the Nrf2 gene. However, the authors are only using the TKO mice to confirm a few things. I was suggesting to use the NEKO mice and test whether the AVP expression levels in the hypothalamus precede or follow the effect on AQ2 levels. This because the reduced AQ2 levels in the kidney could be due to an enhanced vasopressin receptor activation. I was suggesting to use the NEKO mice because this might be easier for them. However, they can use the TKO mice of course, but in this case they must show the levels of circulating AVP and more importantly, the expression levels of AVP in the hypothalamus. In the case they find increased AVP circulating levels and mRNA levels in these mice as well, this would exclude that the effect is not mediated by the lack of Nrf2 in the hypothalamus (because the Cre is specific for the renal tubules) and might indeed suggest that the downregulation of AQ2 in the kidney is secondary to an enhanced AVP and activation on the vasopressin receptor. For the same reason I also was asking to look at the AVP receptor activation state in the kidney and this experiment was also not performed.

Reviewer #4 (Remarks to the Author):

The authors have addressed most of the issues raised in my review and the revised manuscript is significantly improved. The role of the Nrf2 - Clec4 - Aquaporin 2 axis in the phenotype is still based on correlative data, but there is now more evidence for an important role if this axis.

To Reviewer #1:

This Reviewer is satisfied with the Authors' response to the comments raised and finds the study novel and of interest to a wide field of research.

We thank the reviewer for reviewing our paper and also this comment.

To Reviewer #2:

My previous comments related to a better demonstration of the nephrogenic diabetes insipidus state have been addressed by the additional experiments now described in Fig 5 a-c. The supplemental fig 7 demonstrating a normal diluting function is also of interest.

We thank the reviewer for reviewing our paper and also this comment.

Minor: line 197, page 10: dDAVP is a synthetic vasopressin analog, please add analog.

We thank the reviewer for pointing out this oversight. We have added analog to the text.

To Reviewer #3:

The authors have made attempts at answering to most of the questions raised.

We thank the reviewer for reviewing our paper.

However, one major point raised by this reviewer remains non-satisfactorily addressed: the role of AVP levels. This reviewer understands extremely well the importance of confirming the effects observed in the NEKO mice using kidney-specific inactivation of the Nrf2 gene. However, the authors are only using the TKO mice to confirm a few things. I was suggesting to use the NEKO mice and test whether the AVP expression levels in the hypothalamus precede or follow the effect on AQP2 levels. This because the reduced AQP2 levels in the kidney could be due to an enhanced vasopressin receptor activation. I was suggesting to use the NEKO mice because this might be easier for them. However, they can use the TKO mice of course, but in this case they must show the levels of circulating AVP and more importantly, the expression levels of AVP in the hypothalamus. In the case they find increased AVP circulating levels and mRNA levels in these mice as well, this would exclude that the effect is not mediated by the lack of Nrf2 in the hypothalamus (because the Cre is specific for the renal tubules) and might indeed suggest that the downregulation of AQP2 in the kidney is secondary to an enhanced AVP and activation on the vasopressin receptor.

We thank the reviewer for this professional comment. Following this comment, we have conducted a new experiment, and we are happy to report the reviewer that while AQP2 levels are reduced as early as 10 days of age in *Keap1*^{-/-} mice, *Avp* mRNA levels do not change substantially. This observation indicates that the reduction of AQP2 level in the kidney precedes the increase in AVP production, supporting our contention that the increase in AVP levels is a compensatory

response secondary to the AQP2 reduction in the kidney of NEKO mice. We have added this new result of *Avp* mRNA levels in the hypothalamus of *Keap1*^{-/-} mice as Supplementary fig. 4b.

For the same reason I also was asking to look at the AVP receptor activation state in the kidney and this experiment was also not performed.

In order to address this comment, we also have conducted another new experiment. If the reduction of AQP2 level were brought by the activation of AVP receptor, inhibition of AVP receptor would increase the AQP2 level in the kidney of NEKO mice. Therefore, we have treated NEKO and control mice with specific AVP V2 receptor antagonist OPC31260 (OPC; 60 mg/kg body weight intra-peritoneal injection, 4 hour-treatment, n=2), and examined the AQP2 protein level. As shown in the Supplementary figure for the editor and reviewer only, we found that the AQP2 protein level does not increase (or rather decrease) upon the OPC treatment (closed arrowhead, non-glycosylated AQP2; open arrowhead, glycosylated AQP2). This result further supports our conclusion that the increase in AVP is a compensatory response secondary to the AQP2 reduction.

Supplementary figure for the editor and reviewer only

To Reviewer #4:

The authors have addressed most of the issues raised in my review and the revised manuscript is significantly improved. The role of the Nrf2 - Clec4 - Aquaporin 2 axis in the phenotype is still based on correlative data, but there is now more evidence for an important role if this axis.

We thank the reviewer for reviewing our paper and also this comment. We agree that fine mechanistic insights are interesting issues that should be addressed in the near future.

REVIEWERS' COMMENTS:

Reviewer #3 (Remarks to the Author):

The authors have now responded satisfactorily to my requests.